# Amazonian forest-savanna bistability and human impact

Bert Wuyts[1,2], Alan R. Champneys[2] & Joanna I. House[3]

A bimodal distribution of tropical tree cover at intermediate precipitation levels has been presented as evidence of fire-induced bistability. Here we subdivide satellite vegetation data into those from human-unaffected areas and those from regions close to human-cultivated zones. Bimodality is found to be almost absent in the unaffected regions, whereas it is significantly enhanced close to cultivated zones. Assuming higher logging rates closer to cultivated zones and spatial diffusion of fire, our spatiotemporal mathematical model reproduces these patterns. Given a gradient of climatic and edaphic factors, rather than bistability there is a predictable spatial boundary, a Maxwell point, that separates regions where forest and savanna states are naturally selected. While bimodality can hence be explained by anthropogenic edge effects and natural spatial heterogeneity, a narrow range of bimodality remaining in the human-unaffected data indicates that there is still bistability, although on smaller scales than claimed previously.

[1] Bristol Centre for Complexity Sciences, Bristol University, 1-9 Old Park Hill, Bristol BS2 8BB, UK. [2] Department of Engineering Mathematics, Bristol University, Queen's Building, University Walk, Bristol BS8 1TR, UK. [3] School of Geographical Sciences, Bristol University, University Road, Bristol BS8 1SS, UK. Correspondence and requests for materials should be addressed to B.W. (email: bw12158@my.bristol.ac.uk).

International climate negotiations include a focus on reduced greenhouse gas emissions from tropical deforestation and the need for sustainable forest management to enhance sinks. Such forest management is complicated by the potential existence of tipping points[1] in tropical forests beyond which they may experience abrupt transitions[2–4] to savannas and provide feedbacks to climate change[5]. In the Amazon basin, there exists evidence for tipping points related to two types of feedbacks. First, simulation and modelling studies have shown that hydrological feedbacks could cause basin-scale alternative stable vegetation states[6–9]. The focus of this paper is on a second fire-related feedback that has been linked to forest–savanna bistability in the tropics[10–12]. Fire spread requires a spatially well-connected herbaceous layer that occurs only below a certain tree cover threshold; below this threshold, fire spread opens up the canopy more, promoting yet better fire spread[10]. Evidence for this process was found empirically, via higher fire frequencies and a bimodal (see Methods) distribution of tropical tree cover for a certain range of rainfall and seasonality.

A typical characteristic of bistable systems is hysteresis, and in this context it means that the tipping point of rainfall where savannas are converted to forest is higher than that where forests are converted to savannas. In the rainfall range between these two tipping points tree cover would then be observed to have a bimodal distribution because both states are possible here. Simulations of a simple model of bistable tree cover have shown that spatial heterogeneity can enlarge the observed rainfall range of bimodality[13]. This leads to the question of how much of the observed bimodality is due to hysteresis and how much is due to spatial heterogeneity associated with independent variation of other variables that affect stability, such as seasonality, soils and human impact. This question is especially important as previous studies inferred a model of bistability that only focused on the effect of average rainfall[10–15]. If much of the observed bimodality turns out to be due to spatial heterogeneity, hysteresis and bistability may in fact be absent or at least be limited to smaller spatial scales than previously assumed.

Models have reproduced fire-induced forest–savanna bistability by parameterizing or simulating the fire-vegetation feedback[13,15,16], or by only assuming a specific dependence of model parameters on regional climate[17]. Yet, previous theoretical[18,19] studies have shown that when allowing spatial interaction, hysteresis and bimodality disappear; instead, there is an environmentally determined boundary that separates both states. Only at this boundary, coined the Maxwell point (MP), both states coexist. Above the MP, one state dominates while below the MP, the other dominates. Hence, even in the presence of 'local' bistability (that is, when not considering spatial interaction), these models are only 'globally' bistable at the MP (that is, still bistable when considering spatial interaction) (see Methods). Recently, evidence has been found for this phenomenon in satellite data[20] of tree cover. Still, no empirically testable spatiotemporal model including the combined effect of all natural and human influences has been proposed.

In the Amazon region, a gradient of rainfall runs from the dry southeast, where the dry season can have considerable length, to the wet northwest, where dry season is short or nonexistent[7,21] (Supplementary Fig. 1). Natural vegetation follows this climatic gradient. From southeast to northwest, there are dry savannas, moist savannas and eventually tropical forest. Human impact occurs along the same gradient, with drier areas in the southeast having been subject to more land-use change than the wetter northwest. The Amazon region is a good starting point to study human impact effects since the implications of deforestation and logging are well studied there and since there are fewer confounding factors than, for example in Africa, where presence of large herbivores are known to have an important additional effect on vegetation[22].

It has been demonstrated both empirically[4] and with theoretical models[23–25] that human impact can significantly alter the stability and resilience of ecosystems. Human impact in the Amazon region encompasses both direct deforestation and various edge effects around cleared areas such as changes in forest structure, tree mortality, forest microclimate and biodiversity[26]. Deforestation comprises both clearcutting, the conversion of forested land to food crops or pastures and selective logging, the removal of only marketable tree species[27]. Both logged forests and edges of clearcut provide decreased transpiration rates and thus lower atmospheric humidity that, along with scattered wood debris, makes them highly susceptible to fire[28,29]. After being burnt once, nearby forest fragments become yet more susceptible to fire[29]. While previous empirical studies recognize that human impact can influence forest stability, they either focused on bistability in natural systems by excluding affected areas from the analysis[10] or did not explicitly take human impact into account[11].

In this work, we examine human impact on Amazonian forest–savanna bistability. Our key methodology involves three steps. First, we set up a statistical model that predicts pre-human forest cover from average rainfall, rainfall seasonality and soils. Second, we analyse how human impact affects bimodality of tree cover by considering separately areas that are close to and areas that are far from human influence, while using the results from the statistical model in the previous step to remove the confounding influence of natural spatial heterogeneity associated with gradients of climatic and edaphic variables. Third, we derive a spatial stochastic model using observed natural spatial heterogeneity, while also adding edge effects due to deforestation, and compare its output with data. The data analysis shows that without the confounding effect of natural spatial heterogeneity, substantial bimodality is only observed for places close to agriculture. The model results indicate that the bimodality close to agricultural zones can be explained by anthropogenic edge effects due to logging and fire spread. Model results further show a sharp boundary between savanna and forest at the MP point, predictable from climate, soils and distance from human impact. This shows that hysteresis is not required to reproduce bimodality. However, some limited remaining bimodality after accounting for natural and anthropogenic spatial heterogeneity indicates that there are regions of global bistability, although on smaller scales than previously recognized.

## Results

**Predicting pre-human forest cover from climate and soils.** Many of the areas that have been savannas for a long time are colonized by humans. Restricting our analysis to pristine areas in deriving relations between natural variables and forest would then possibly lead to biased estimates of natural effects or an under-estimation of hysteresis in the system, if present. Therefore, we necessarily start from an estimate of pre-human forest cover that we take from the World Conservation Monitoring Centre (WCMC) original cover data set[30].

Figure 1a shows the WCMC data of pre-human forest cover compared with current forest cover, obtained from the MODIS (Moderate-resolution Imaging Spectroradiometer) vegetation continuous field (VCF) data set[31]. Comparing Fig. 1a with Fig. 2b, one can see that forest areas that have been lost through deforestation almost exclusively occur in or around agricultural areas. We performed a logistic regression model on these data, predicting pre-human forest cover from mean annual rainfall (MAR in the text, $P$ in the equations), Markham seasonality index (MSI in the text, $M$ in the equations), topsoil bulk density ($\rho$),

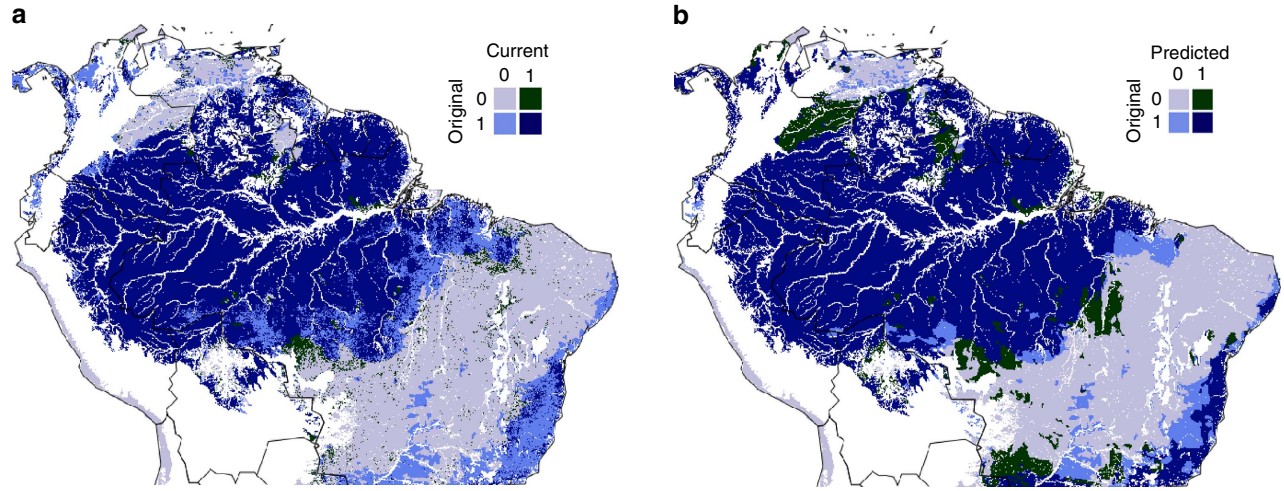

**Figure 1 | Maps of observed and predicted pre-human forest cover.** (**a**) Pre-human forest cover (WCMC Original Forest Cover Map[30]) compared with current forest cover (MODIS vegetation continuous fields[31]). (**b**) Predicted pre-human forest cover with the logistic regression model shown in Table 1 compared with observed pre-human forest cover. See Supplementary Fig. 2 and Supplementary Table 1 for alternative prediction models.

**Table 1 | Logistic regression model and parameters.**

| $g(F(\mathbf{y})) = \theta_0 + \theta_1\varphi_s + \theta_2\varphi_c + \theta_3\rho + \theta_4\varphi_c\rho + \theta_5M + \theta_6P,$ | | | |
| --- | --- | --- | --- |
| $i$ | | $\theta_i$ | $\theta_i$ (standardized) |
| 0 | Intercept | 4.16 | 1.80 |
| 1 | Sand | 2.38 e − 02 | 6.08 e-01 |
| 2 | Clay | − 1.88 e − 01 | 1.10 |
| 3 | Density | − 5.99 | − 9.09 e − 02 |
| 4 | Clay/density | 1.83 e − 01 | 4.79 e − 01 |
| 5 | MSI | − 7.05 | − 1.09 |
| 6 | MAR | 2.90 e − 03 | 2.29 |

Model fitting was done on 50,000 samples (1 km spatial resolution). All parameters have $P$ values $< 1e - 04$. Forest is predicted where the log odds of forest occurrence $g(F(\mathbf{y})) > 0$ (see Methods for details). The κ-agreement index with data is 0.69 (substantial agreement). See Supplementary Fig. 2 and Supplementary Table 1 for alternative prediction models.

topsoil sand fraction ($\varphi_s$) and topsoil clay fraction ($\varphi_c$). The climatic data were obtained from the TRMM merged satellite-gauge rainfall data set[32] and the soil data from the harmonized world soil database[33]. The regression equation for the log odds of forest occurrence is

$$g(F(\mathbf{y})) = \theta_0 + \theta_1\varphi_s + \theta_2\varphi_c + \theta_3\rho + \theta_4\varphi_c\rho + \theta_5M + \theta_6P, \quad (1)$$

with coefficients shown in Table 1. The graphical representation of this equation is the surface in the space of predictor variables that best separates forest from nonforest, also called the decision boundary[34]. The coefficients in Table 1 can be seen as the components of a vector perpendicular to the decision boundary; the larger a particular component of that vector, the greater the influence of the corresponding predictor on occurrence of forest. The largest effects are a positive effect of MAR, a negative effect of MSI and a mostly positive effect of soil clay fraction. Using either MAR or MSI on their own fails to predict forest cover satisfactorily (Supplementary Fig. 2c,d and Supplementary Table 1). Only when considering their combined effect (Supplementary Fig. 2b), is forest cover predicted well. Further taking into account soils leads to a better prediction of the Atlantic Forest (Supplementary Fig. 2a versus Supplementary Fig. 2b). This suggests that moist forests can only exist there due to favourable soil conditions. The interaction term of density and topsoil clay fraction implies that the effect of clay depends on density or the other way round. If we assume the former, the effect of clay is positive except for exceptionally low-density soils

($<5\%$ density percentile). A positive effect of clay fraction is consistent with previous empirical studies[35]. There is also a separate negative effect of bulk density that is consistent with the effects of higher soil compaction at higher densities. The more complicated effect of soils is most likely a consequence of the nonlinear relation between soil texture and soil hydrology.

**Current tree cover in different human impact zones.** In this section we show scatterplots of current tree cover similar to previous studies[10,11], using 2010 MODIS VCF percent tree cover data[31] (MOD44B collection 051) on a 250 m spatial resolution. Comparing the scatterplots with points sampled from areas close to human impact versus far from human impact will reveal how humans affect the dynamics. The spatial distribution of forest cover is shown in Fig. 2a. Figure 2b shows our subdivision of the study area in human impact classes. We assume edge effects operate up to a distance of 3 km from agricultural or urban areas[26] and have termed this the 'transition' zone. Areas $>3$ km are deemed 'natural'. For this classification, publicly available satellite data[36,37] (see Methods for details) were utilized.

Figure 3 shows a plot of current tree cover sampled from natural areas versus MAR (Fig. 3a) and versus MSI (Fig. 3b). Confirming previous studies, there is a rainfall range of bimodality, here between 1,400 and 1,900 mm where dense tree-covered forests and sparse tree-covered savannas both exist (Fig. 3a). However, there exists a significant difference between the MSI (colour scale) of forest and savanna, indicating that at

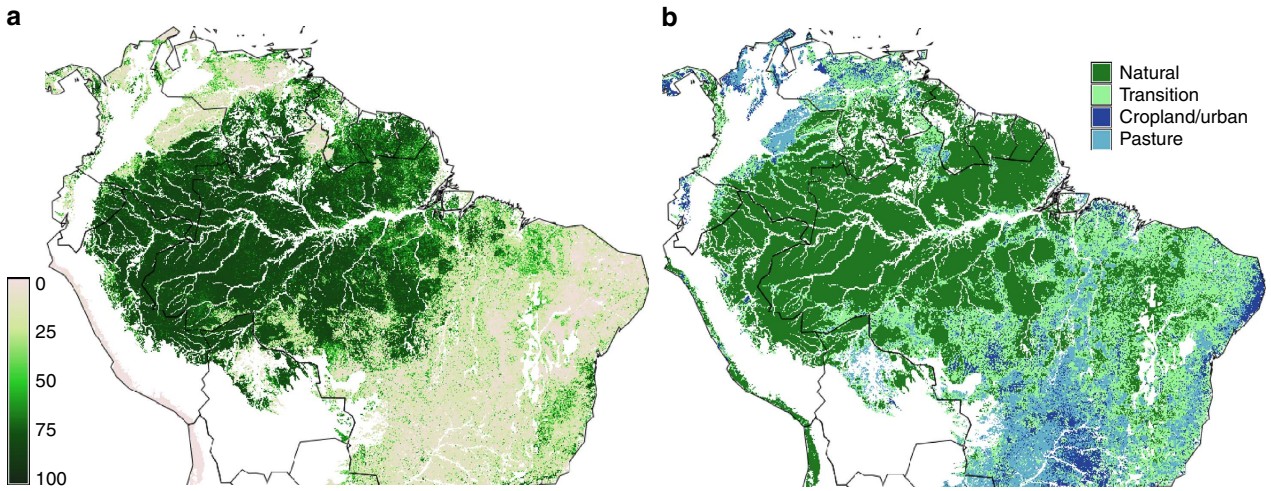

**Figure 2 | Study area with tree cover and human impact classes. (a)** Map of MODIS VCF tree cover in 2010. (**b**) Human impact classes (colours), excluded areas (white).

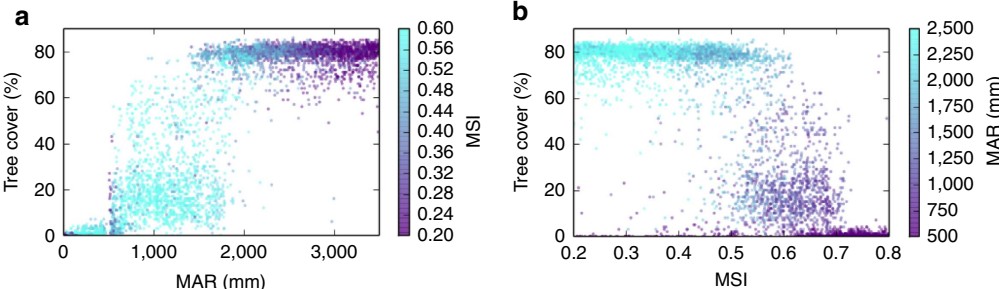

**Figure 3 | Scatterplot of tree cover in natural areas. (a)** Tree cover versus (and stratified on) mean annual rainfall (MAR) with Markham seasonality index (MSI) as colour scale and (**b**) Tree cover versus (and stratified on) Markham's seasonality index with MAR as colour scale. In the rainfall range of bimodality (1,400–1,900 mm), the differences of seasonality between forest (>40% tree cover) and savanna (<40% tree cover) were found significant with a t-test and also with the nonparametetric Mann–Whitney test ($P \approx 0$). This is also the case for the rainfall differences in the bimodality range of seasonality (0.45–0.62) ($P \approx 0$).

least some of the bimodality may be due to spatial heterogeneity associated with effects of seasonality that are independent from those of rainfall. The scatterplot versus MSI (Fig. 3b) shows similarly that in the MSI range of bimodality, there is a significant difference between the MAR (colour scale) of forest and savanna.

To properly visualize the anthropogenic effects on bimodality, we need a measure that combines all predictor variables while minimizing the confounding effect of natural spatial heterogeneity. The best measure for this is the one that quantifies the distance perpendicular to the decision boundary. This is exactly what is done by the expression shown in equation (1). As it is also a measure for the suitability of the natural environment for moist forest, we refer to it further as the climatic–edaphic forest suitability (CEFS). Places with a large negative value are naturally unsuitable and places with a large positive value are suitable. If the human-unaffected system exhibits hysteresis on large scales, as suggested by previous work, we should see a wide interval of bimodality in a scatterplot of current tree cover versus CEFS where points are sampled from human-unaffected areas. In the case of absence of hysteresis, CEFS = 0 would separate low and high tree cover in such a scatterplot, and there would be no bimodality.

In Fig. 4 we show the scatterplots, using the 2010 VCF data versus CEFS (stratified on CEFS), sampling separately from areas close by and far from human impact. The scatterplots also indicate via a colour scale the median of the tree cover change

between 2000 and 2010. When looking across the entire nonagricultural area (natural + transition class, Fig. 4a) the observations confirm previous findings[10,11]; with overlapping forest (defined here as >40% tree cover) and savanna (5–40%) states for CEFS in the range from about −0.9 to 2.5. However, when looking at the areas distant from human influence (Fig. 4b) we find less overlap of savanna and forest states (−0.9 to 0.8), with a sharp front around CEFS = 0. For the transition areas (Fig. 4c), the range where the two states overlap is much wider (−0.9 to 5). We cannot verify based on this analysis whether the treeless state (<5% tree cover) occurs as a discontinuous transition from the savanna state (Fig. 4b) as CEFS is the optimal combination of predictors for the savanna–forest transition and not for the treeless–savanna transition. Besides, there has been criticism about inferring such properties over short tree cover ranges from the MODIS VCF data[38–40].

Looking at the colour scale in Fig. 4, we can draw tentative conclusions about the underlying causes of tree cover change. The increasing effects of fire and drought with decreasing CEFS (see below) mean that below a CEFS = 0, high/intermediate tree cover areas experience considerable losses. Natural areas with positive CEFS experienced a tree cover increase between 2000 and 2010 with median values of up to 20% in 10 years (Fig. 4b). This is presumably due to recovery from natural disturbance or a response to environmental change. In transition areas, dense forests experience losses at all CEFS values (Fig. 4c).

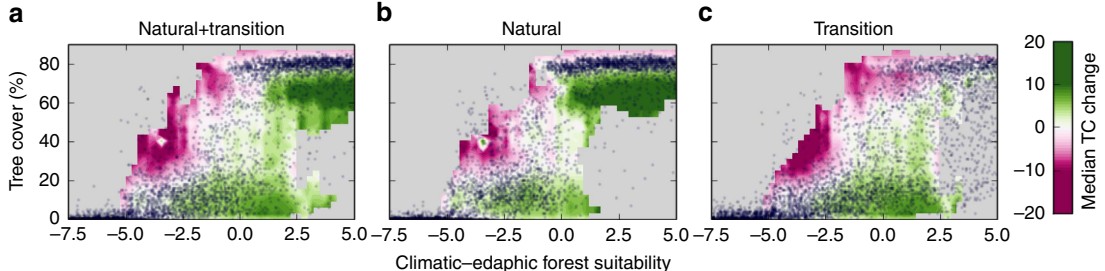

**Figure 4 | Scatterplots of tree cover versus (and stratified on) CEFS.** (**a**) Natural + transition areas. (**b**) Natural areas. (**c**) Transition areas. The colour scale indicates percent change in cover between 2000 and 2010 as a function of tree cover of 2000 and climatic–edaphic forest suitability, with green denoting increase in tree cover, purple decrease and grey regions with insufficient data (<200 pixels).

To complement these data, we have also looked at the AQM (área queimada; burnt area in Portuguese) fire occurrence data[41] in the study area. Observed fire frequencies are lower at the very lowest and at the higher end of tree cover, with a hump in the middle (Supplementary Fig. 4b). This is consistent with the intermediate fire-productivity hypothesis[42]. At the low end, in dry treeless areas (<5% tree cover), which occur mainly in the Sechura–Atacama deserts along the Peruvian and Chilean coastline, grass occurrence is very low[21]. Hence here, the lack of flammable material prevents fire occurrence or spread. At intermediate tree cover, in savannas, fires are not limited by fuel but by drought frequency. At high tree cover, droughts are less frequent, allowing the canopy to close at cost of the flammable grassy layer, resulting in lower fire frequencies. We further produced a relative histogram of fire frequencies as a function of tree cover and annual water deficits to see how dryness affects fire frequency (Supplementary Fig. 3). The tree cover value below which fire occurrence becomes important increases steadily with dryness. Fire frequencies are highest in savanna areas, with tree cover of ∼5–40% and annual water deficits of ∼400–800 mm (Supplementary Fig. 3). Above 800 mm, tree cover is very low while fire frequencies are low too, indicating that water limitation is affecting trees severely here.

**Mathematical model**. To explain these findings, we set up a stochastic partial differential equation model[43], inspired by the ordinary differential equation model in ref. 15, for the effect of fire on the following cover types: forest tree, savanna tree, savanna sapling, grass and bare soil. The model has as natural external variables the spatial distribution of observed MAR ($P$) and edaphic suitability for forest ($\pi$). Deforestation is taken into account by having a forest removal term that decays exponentially with distance to the agricultural/urban class ($\delta$). These external variables influence the dynamics by affecting growth and mortality rates in the equations. In modelling fire, we include both local fire and fire spread between pixels (see Methods section for more detail). The humped shape of fire frequency versus tree cover also arises in the model, leading to a good match between fire rate data and modelled fire rates (Supplementary Fig. 4b). In the model, this shape is caused by the high fraction of bare pixels in treeless areas, reducing the amount of fire-prone material at low tree cover, and the assumed reduced fire occurrence at high tree cover (above ∼40% tree cover).

The results are plotted in Fig. 5e–h and show the necessity of both logging and fire diffusion from agricultural areas to reproduce the patterns seen in the observations (Fig. 5a-d; see also Supplementary Movies 1 and 2). Inclusion of these effects results in little overlap between the states for natural areas, just as in the data (Fig. 5c,g). Logging alone (Fig. 5l) or fire alone (Fig. 5p) is insufficient to cause forest–savanna shifts in the transition region similar to the data (Fig. 5d). Note that in our

model logging is not necessary to have fires creeping into the forest. Anywhere along the front, independently of whether it is determined by climate/soils or by human impact, fires will spread into the forest. As in the data, the treeless state mainly occurs along the arid Peruvian–Chilean coastline. In the data, there is much more noise in the range of CEFS where savannas occur (−5 to 0) and some remaining overlap around CEFS = 0.

To explain the model findings, we created a phase diagram by running the model on a 50 × 50 lattice, starting from random initial conditions, for all relevant combinations of $\delta$ and $P$ (Fig. 6a). Figure 6b shows the prediction of these states in space based on this diagram. Only around the point where savanna and forest are equally stable, that is, the MP, do forest and savanna coexist (Fig. 6a,e). Places that are drier or closer to human-affected zones end up in a homogeneous savanna (or treeless) state (Fig. 6a,d), while places that are wetter or further from human impact end up in a homogeneous forest state (Fig. 6a,f). The almost perfect agreement with the model run on the whole study area (Fig. 5e) indicates that forest, savanna and treeless states can be predicted based on $P$, $M$, $\delta$ and $\pi$.

## Discussion

The data analysis shows that after accounting for the most relevant sources of natural spatial heterogeneity, there is still bimodality between Amazonian forest and savanna states, but it is less extensive than previously thought and largely restricted to the transition regions within 3 km of urban/agricultural land. The model shows that much of the Amazonian bimodality in transition regions can be explained by anthropogenic edge effects involving increasing forest removal towards agricultural areas and resulting increases of fire occurrence and spread. Hence, for the Amazon region, the data and the model support the notion that bimodality in rainfall-tree cover plots of the spatially aggregated data in previous research[10–12] is mostly due to spatial heterogeneity[13] associated with rainfall seasonality, soils and human impact, and not due to a large-scale hysteresis. Instead, as in theory of bistable systems with sufficient spatial interaction[19,20], there are two spatially distinct zones of savanna and forest states with their boundary occurring at the MP. The location of the MP is predictably dependent on climate, soils and distance to human impact. Bimodality over large ranges of predictor variables as found in previous studies arises hence because they did not consider the joint effect of all relevant variables but focused only on that of MAR (see Fig. 3), or anthropogenic edge effects are ignored when the data are spatially lumped (see Supplementary Fig. 6). Furthermore, the bimodality found in previous studies was consistent with models of bistability because the models did not include spatial interaction that allows fire to seep into forested areas. Even in spatially explicit percolation models[16], such spilling effects at the

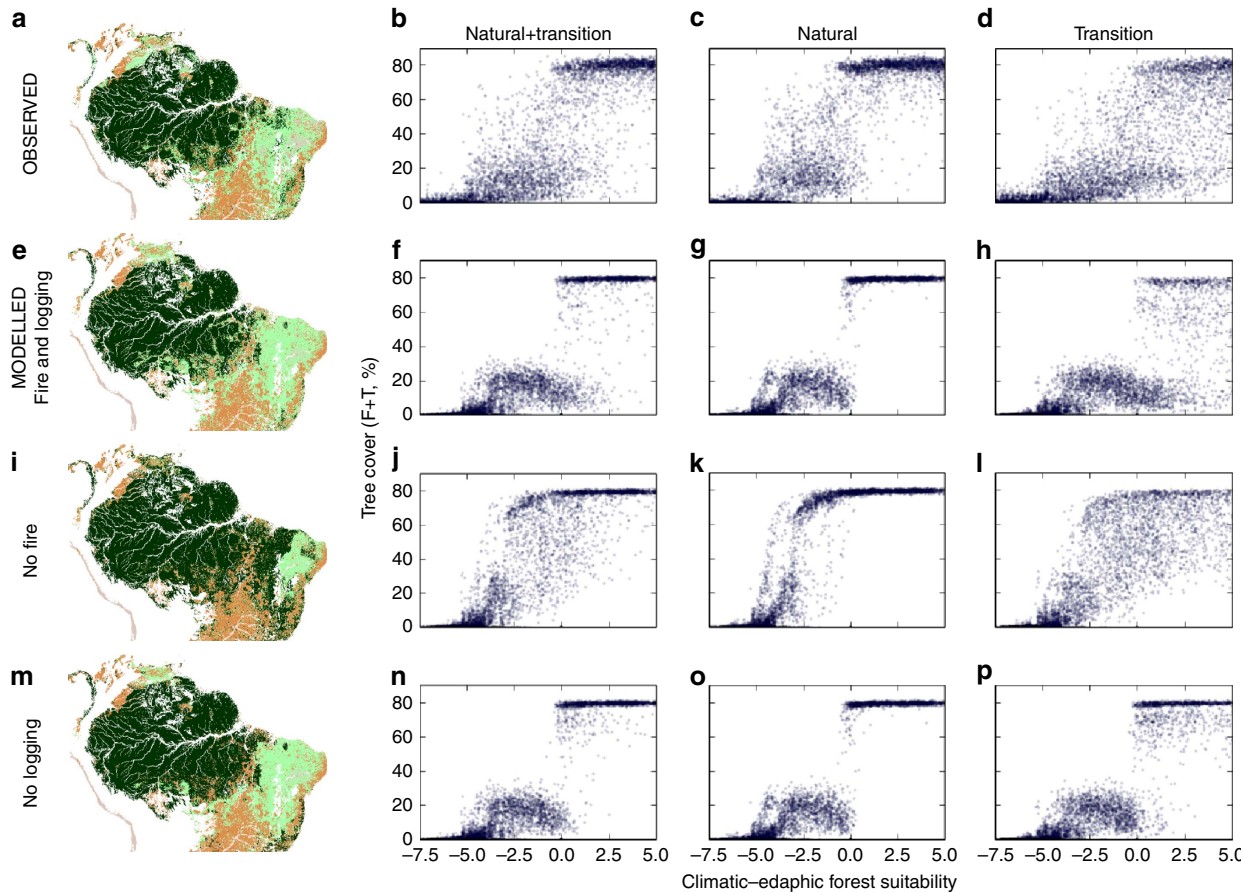

**Figure 5 | Model results with the effects of different terms compared with data via maps of tree cover classes and scatterplots of tree cover versus climatic–edaphic forest suitability.** (a–d) Observed tree cover in 2010. (e–p) Modelled tree cover: (e–h) fire and logging, (i–l) no fire and (m–p) no logging. Respective κ-agreement indices of the modelled forest state with the data are 0.73, 0.42 and 0.69. See Supplementary Fig. 5 for differentiation of these results with respect to cover type. Forest (40–100%): dark green; savanna (5–40%): light green; treeless (<5%): grey; agricultural/urban areas: beige.

forest–savanna boundary were not conceptualized, as fire only percolates on flammable clusters, to which forest does not belong.

However, the small amount of remaining bimodality in the data for natural areas around the MP shows that a role of hysteresis on smaller scales cannot be ruled out. If the system truly does not have hysteresis, as in the model, the remaining overlap could be due to additional unmodelled spatial heterogeneity, such as topography, radiation, temperature, variation of plant physiology, fire/drought resistance of forest and natural edge effects or spatiotemporal stochasticity from, for example, interannual climate variability or longer-range fire spread. Among the effects of spatial heterogeneity, we expect that fire/drought resistance of forest may play a relatively large role. Even though we only considered moist lowland forests by excluding dry and montane forests from our analysis, different adaptations of moist lowland forests to drought and fire may still explain small but discernible differences in the MP. Furthermore, an important next step in validating spatial forest–savanna models is the explanation of the distribution of dry forests based on an understanding of their adaptations to drought and fire. Natural edge effects related to microclimatic differences and increased windthrow[44] may explain some of the remaining overlap. As most edge effects occur within a couple of hundreds of metres from the edge[26], the overlap they cause is expected to be much smaller than that caused by logging. The stochastic effect of interannual climate variability[45,46] and of the occurrence of relatively rare but large fires[47] would blur the MP into a small

overlap region, where both savanna and forest can occur. If we did in this study account sufficiently for the effects of heterogeneity and stochasticity, the remaining overlap is due to bistability and hysteresis on a smaller scale than previously recognized. One possible explanation for hysteresis is the existence of vegetation–climate feedbacks[6–9]. Earlier studies based on simple nonspatial models have shown that such feedbacks could amplify the hysteresis associated with feedbacks on smaller scales[13,48].

If, as assumed in previous studies, the whole rainfall range of bimodality were due to hysteresis[10–12,15], there would be two tipping points—one at low rainfall for forest–savanna transitions and another at high rainfall for savanna–forest transitions. Forests would only recover naturally after rainfall has increased beyond the upper tipping point. Our results suggest that such large rainfall increases are not necessary for forest recovery since there is limited hysteresis, with both tipping points close to the MP. Consistent with the phase diagram in Fig. 6a, the MP will shift as agriculture expands. Naturally deterministic forests near to these expansions will turn into deterministic savannas closest to agricultural zones and into globally bistable (see Methods) areas a bit further away, up to the point where anthropogenic edge effects decay to zero. Similarly, naturally bistable forests in close proximity of agricultural expansions will turn into deterministic savannas. However, as the human effects operate on a much smaller scale than the natural effects do, the zones of global bistability around agricultural areas will be much more

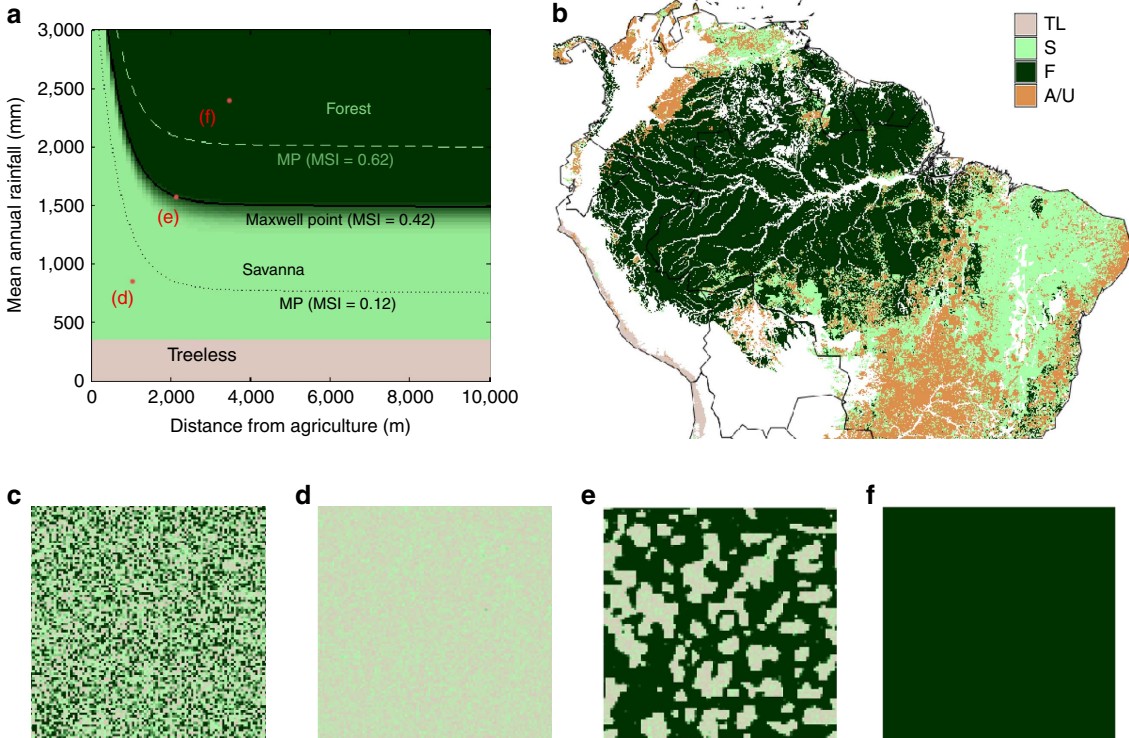

**Figure 6 | Model states predicted from steady states at the end of simulations on a square lattice for all relevant combinations of MAR and distance to agriculture.** (**a**) State diagram as function of distance from agriculture ($\delta$) and MAR ($P$) (example with average seasonality and soil conditions). Between savanna and forest, the colour scale represents the proportion of the lattice that is occupied by adult trees. The solid black line shows where the Maxwell point lies, that is, where this proportion settles to a stationary value that is not 0 or 100%. The dotted line shows the Maxwell point for high seasonality (MSI = 0.86, 95% percentile) and the dashed line for low seasonality (MSI = 0.12, 5% percentile). (**b**) Classification of the study area using $P$, $M$, $\delta$ and $\pi$. (**c–f**) Example simulations: (**c**) initial condition, (**d–f**) final states with $\delta$ and $P$ values at the points indicated in **a**.

narrow than the globally bistable area due to the natural gradients. Therefore, in naturally deterministic areas, changes to land management practices around human-impacted areas would be sufficient to enable forest recovery. In the areas that were globally bistable before human expansion, a slight improvement of climatic conditions may be necessary in addition. Theory suggests that the front speed during forest recovery will be a function of how close the forest's conditions are to the natural MP[18,19,49]. The more the climatic and edaphic favourability of a secondary forest is beyond that given by the MP, the faster the recovery will occur. This is consistent with the climate-dependent resilience found in recent research on secondary forest recovery[50].

Notwithstanding the remaining uncertainty about the existence of hysteresis on a smaller spatial scale, we have shown that most of the previously observed bimodality of tree cover in the Amazon basin can be explained by natural and anthropogenic spatial heterogeneity. This was overlooked in previous studies because they ignored space. In their data analysis, spatial heterogeneity of other variables than rainfall was not taken into account, and in their models, spatial interaction was not captured. As this led to large ranges of bimodality in the data and large hysteresis loops in the model, data and models agreed not due to being valid but due to biases that worked in the same direction for both. A greater understanding will now be required of how other important processes and phenomena such as longer-range fire spread effects, hydrological feedbacks[6–9], regional adaptations of forests to drought/fire, interannual climate variability[45,46] and climate change may cause or affect[48] local hysteresis, and how they influence the MP. The effects of certain feedbacks may not be inferable from a merely spatial analysis or a limited temporal record. Therefore, hydrological feedbacks may still cause basin-scale hysteresis associated with vegetation–climate interaction. Nonetheless, both the local-scale fire feedback and the regional-scale hydrological feedback exist due to spatial interactions that—as we have shown—can be crucial to the dynamics and therefore should not be ignored in future models of tropical vegetation, whether conceptual or for simulation.

## Methods

**Study area.** We delineate our study area with geographical coordinates 12.5°N, −23°S, −81.5°W and −34.5°W. All analyses of spatial data have been restricted to the land area contained within the bounds of this rectangle. As in ref. 10, we exclude all areas above 1,500 m altitude (with the SRTM DTM[51] data set) because other effects than rainfall (for example, temperature) may become important there. Additionally, places above 1,500 m altitude generally have complex topography and microclimate such that interpolation errors are greatest in these regions. We have also excluded wetlands and areas around major rivers, since tree cover may be affected in various ways there due to permanent high water availability, flooding and human impact, and may introduce more noise, although these areas are spatially relatively restricted. We have further excluded areas where forests may have specific or regionally distinct adaptations to drought, fire or altitude. These include all montane and dry forests as defined by the WWF ecoregion database[52].

**Observed tree cover.** Tree cover data was obtained from the satellite-derived MODIS continuous vegetation fields data set[31]. Areas where at least three out of seven periods used for reconstruction of tree cover in 2000 or in 2010 were affected by cloudiness or satellite-related issues (Supplementary Fig. 7) were excluded from the analysis to mitigate the influence systematic errors. Systematic errors tend to depress tree cover estimates due to the lowering effect of cloudiness on normalized difference vegetation index. We expect random errors to be reduced by using the median tree cover change as a robust population estimate of the change between 2000 and 2010 in Fig. 4.

Recently, the MODIS VCF data set has received some criticism[38] since the method used to estimate tree cover from remote-sensed data (classification and regression tree (CART) analysis) can introduce artificial discontinuities and hence multimodality in the data, even if the underlying tree cover data are uniform. This

indicates the importance of using data sets that use alternative methods to produce tree cover estimates when testing the multistability hypothesis, particularly since CART methods have been widely used in mapping ecological variables based on remote-sensed data. Nevertheless, a follow-up correspondence explored the error distribution in the MOD44B data set in more detail using local ground measurements to show that the previously found bimodality of tree cover is not attributable to bias in the calibrations[39]. They also concluded that the data are not well resolved below 20–30% tree cover and may not be useful for comparing over ranges smaller than ∼10% tree cover. While the critics acknowledge the use of the VCF data to detect course patterns, they maintain that we should be wary of potential biases produced by CART methods[40]. Hence, we cannot deduce from these data whether the treeless state is a continuous or discontinuous transition. Nevertheless, in our work we mainly focus on forest–savanna bistability and therefore our conclusions are expected to be unsensitive to this issue.

**Pre-human forest cover.** The United Nations Environment Programme–World Conservation Monitoring Centre (UNEP-WCMC) data of pre-human forest cover[30] is the result of a combination of potential forest cover maps, satellite data and expert knowledge on ecoregions. In the source material it is mentioned that the data represent forest cover before the arrival of the 'modern man'. However, even before colonization, advanced societies existed. For instance, in South America, sophisticated knowledge of land management methods by fire[53] caused large impacts on the environment. The size of these societies was reduced with 90% by disease due to first contact with European colonists, followed by a substantial decrease in fire activity, as evidenced by frequency of charcoal particles in paleoenvironmental records[53]. An estimation of human impact during this transition[54] shows that it took two centuries for human impact to recover up to the same level again. If precolonial human impact would have created savannas, this amount of time was likely enough for forest to fully recover again. We assume therefore that the data represent forest cover before emergence of methods that have a substantive impact on the environment, whether right before the arrival of modern man or somewhat earlier.

**Soil data.** We took the data on top soil (top 30 cm) sand fraction, clay fraction and bulk density from the Harmonized World Soil Database[33] and used them as predictors of pre-human forest and as variables in the model. See Supplementary Fig. 1c–e for their maps. Note though that the quality of these data are lower than that of the climate and vegetation data.

**Fire data.** We chose the AQM burnt area data set of INPE (Instituto Nacional de Pesquisas Espaciais)[41] and not the commonly used MODIS burnt area data since the former has better agreement with high-resolution LANDSAT data on burnt area, with a lower number of omission errors, and was calibrated to work specifically for Brazil instead of globally.

**Climate data and processing.** MAR and MSI[55] were derived from the 0.25° merged gauge-remote-sensed data of the Tropical Rainfall Measuring Mission (TRMM 3B43)[32], with a temporal coverage from 1998 to 2010. Merging remote-sensed and gauge data balances their respective errors caused by biases and the inhomogeneous distribution of weather stations respectively. MAR and MSI are defined as

$$\text{MAR} = 1/13 \sum_{y=1998}^{2010} \sum_{m=1}^{12} p_{y,m},$$

$$\text{MSI} = \frac{\left\| \sum_{m=1}^{12} \mathbf{p}_m \right\|}{\sum_{m=1}^{12} p_m}, \qquad (2)$$

where $p_{y,m}$ represents rainfall in month $m$ of year $y$ and $\mathbf{p}_m$ a vector with magnitude the multi-year monthly average of the $m$th month in the year and as phase $m\pi/6$. MSI varies between 0 and 1, with higher values meaning higher seasonality. For maps of MAR and MSI, see Supplementary Fig. 1a,b.

It was found previously that both MAR and mean dry season length correlate with tree cover[10]. Therefore, scatterplots that serve to find parameter ranges where bistability is likely to occur should either be plotted as function of both MAR and mean dry season length or of one variable that captures both effects. We used mean CEFS for this purpose that additionally takes into account the effect of soils (see main text). To see how all used climatic variables in this text correlate, see Supplementary Fig. 8.

**Prediction of pre-human forest cover from natural variables.** We performed a logistic regression on the above-mentioned data of estimated pre-human forest cover to test which variables significantly affect natural forest cover, justify the use of them in our model and to be able to construct an optimal combination of predictor variables that captures all relevant predictor information in one variable. The regression was done on a random sample of 50,000 points on a spatial resolution of 1 km. We performed ridge logistic regression to assure that dependence between the predictors was not an issue. We exclude highlands ( > 1,500 m),

montane/drought-adapted forests and wetlands. The used climatic variables are mean annual rainfall ($P$) and the Markham seasonality index ($M$). Used edaphic variables are topsoil sand ($\varphi_s$) and clay ($\varphi_c$) fraction, and topsoil bulk density ($\rho$). The dichotomous response variable is forest occurrence. The logistic regression was used to classify places as forest or nonforest via demarcation of a decision boundary in the space of predictor variables, where the log odds equals zero, meaning that the probability of forest occurrence equals 50%. Forest is predicted beyond this boundary, where the log odds is greater than zero, or,

$$g(F(\mathbf{y})) = \ln[F/(1-F)] = \boldsymbol{\theta}.\mathbf{y} > 0, \qquad (3)$$

with $F(\mathbf{y}) = [1 + \exp(-\boldsymbol{\theta}.\mathbf{y})]^{-1}$ the probability of forest occurrence given $\mathbf{y}$, where $\boldsymbol{\theta}$ is the vector $(\theta_0, \theta_1, \theta_2, \theta_3, \theta_4, \theta_5)$ and $\mathbf{y}$ the vector $(1, \varphi_s, \varphi_c, \rho, \varphi_c, \rho, \mu)^T$. In ref. 11, logistic regression has been applied using mean annual rainfall instead of CEFS, and without taking into account human impact, data quality and altitude/drought adaptations.

**Tree cover versus CEFS scatterplots.** The scatterplots of remote sensed tree cover (250 m spatial resolution) against CEFS were made at tree cover resolution (as in refs 10,11) by generating high-resolution maps of CEFS via spline interpolation. Due to the high number of pixels in the study area on this resolution, we chose to make scatterplots from a random spatial subsample of the data set, stratified by CEFS. We did this by taking 100 samples out of 50 consecutive CEFS intervals between − 7.5 and 5. The tree cover (TC) data are available for 11 years (2000–2010), and therefore we can also analyse TC stability based on its dynamics rather than just its spatial distribution. By binning tree cover change between the years 2000 and 2010 in two-dimensional bins of [CEFS,TC(2000)] space [with bin width (CEFS;TC) = (0.25; 3%)], we reconstructed the net growth as function of CEFS and TC by calculating the median in each of the bins. This analysis was done for each of the human impact classes. It has to be noted that regions with CEFS lower than about − 5 occur far less in the study area (Supplementary Fig. 1f). Above this value of CEFS, the observed frequencies in the different bins of CEFS are fairly comparable. Hence, in those drier places with CEFS lower than about − 5, which occur almost exclusively in the Sechura and Atacama desert ecoregions in Western Peru and Chile, sampling will be denser and therefore prone to biases. Nonetheless, here we almost exclusively focus on the forest–savanna front that occurs around CEFS = 0. Scatterplots of tree cover versus MAR/MSI were produced in the same way.

**Subdivision into classes of degree of human impact.** Based on the rationale that human impact decreases with distance from agricultural and urban areas, we subdivided the study areas into different classes of human impact. We used two data sets for this purpose: MODIS land cover [IGBP (International Geosphere-Biosphere Programme) classification][37] (500 m resolution) and the pasture intensity variable of the Agricultural Lands data set[36] that is the result of downscaling subnational statistics on a 10 km grid via remote-sensed land cover data. The agricultural and urban class in Fig. 2b contains those areas that consist of cropland (land cover classes 12 and 14), have a pasture intensity higher than 0.5 or are urban (land cover class 13). The transition class contains the areas that lie not further than 3 km from these agricultural/urban areas. The natural class consists of areas further than 3 km from the agricultural/urban class. We chose the value of 3 km as it is slightly higher than the value of 2.4 km found for the largest edge effects[26].

The data on pasture intensity were interpolated to 250 m resolution by subdividing every 10 km$^2$ pixel into 1,600 250 m$^2$ subpixels and allocating the intensities proportionally to $H_{i,j}/H_j$, where $H_{i,j}$ stands for herbaceous cover in the $i$th subpixel of pixel $j$. The data on herbaceous cover were taken from the MODIS VCF data[31].

**Separating forest and savanna.** We chose 40% as threshold to separate forest from savanna based on analyses of fire occurrence versus tree cover. Supplementary Fig. 3 shows a plot of observed fire occurrence in the study area as a function of tree cover and annual water deficit. This plot is based on annual fire and tree cover data from between 2005 and 2014 (in nonimpacted places). It shows that the tree cover value where fire occurrence increases sharply increases with increasing dryness. That the fire feedback occurs from higher tree cover values in drier areas should have an effect on which tree cover values are observed at the forest side of the forest–savanna transition. Nonetheless, most burning occurs below about 40% as shown in Supplementary Fig. 3. Likewise, due to the fire–vegetation feedback, there are very few areas with tree cover between 40 and 60% compared with outside that range. Therefore, the actual choice of the threshold in this range does not really matter much.

Previous studies have also found 40% as the threshold based on data analysis[12,56]. The 40% result is also consistent with percolation theory on two-dimensional (2D) square lattices. The problem of fire spread in grasses is equivalent to what is called 'site percolation' on 2D square lattices with nearest neigbours neighbourhood, for which a percolation threshold of 59% has been found[57,58]. The 59% refers to the density of the medium over which the percolation occurs—in our case the grass layer. Therefore, the tree cover value is 100 − 59% = 41%. The percolation threshold generally depends on the type of percolation ('site' versus

'bond'), the type of lattice (square, triangular, honeycomb and so on), its dimension (1D, 2D, 3D) and its neighbourhood structure (only nearest neighbours, nearest neighbours and next nearest neighbours and so on). That the threshold in savannas is close to the one for nearest neighbour 2D site percolation suggests that on average, patches of grass in savannas are connected as nodes in a square lattice. The lowering of the grass density threshold with dryness (Supplementary Fig. 3) suggests that the effective connectivity is increased by dryness.

Furthermore, there is also the question of whether low tree cover ($<40\%$) really represents savanna and high tree ($>40\%$) cover forest. Savanna trees rarely grow higher than 15 m while forests reach canopy heights of $\sim 40$ m. This was addressed in a recent study[59], where they compared the VCF tree cover data set with satellite data on canopy height. The high agreement of the low cover with the low canopy height and the high cover with the high canopy height mode confirmed that the VCF data separates savannas and forests well. However, some low cover areas were found to have high canopies, indicating degraded forests. Using the same data set, we verified where these high height low cover areas ($>15$ m and $<40\%$) lie and found them to be almost exclusively in or close to transition areas (Supplementary Fig. 9). This suggests that our transition class indeed captures anthropogenic degradation.

**Bimodality and local versus global bistability.** We use the term bimodality when seeing both forest and savanna states exist at the same rainfall/seasonality/climate–edaphic forest suitability. Bistability and hysteresis are system-dynamic properties while bimodality only refers to the statistical distribution of a variable, tree cover in this case. We further use the term local bistability for existence of alternative stable states for a parameter range in models without spatial interaction. If there are still alternative stable states for a certain parameter range despite having spatial interaction, we use the term global bistability. When only talking about bistability, we refer to global bistability, as this is what would be observed in nature.

**Model.** To explain the findings from the data analysis, we set up a stochastic[43] partial differential equation model inspired by ref. [15] for the effect of fire ($\Phi$) on the following cover types: forest tree ($F$), savanna tree ($T$), savanna sapling ($S$), grass ($G$) and bare soil ($B$) depending on space $x$ and time $t$, expressed as fraction of space occupied,

$$\partial_t S = g_S(M,P)GT - q(\Phi)S - m_S(M,P)S - g_F(M,P)SF - r_0 S\mathbf{1}_{(\Xi,\Theta)} + \sigma G\xi_S,$$
$$\partial_t T = q(\Phi)S - m_T(M,P)T - g_F(M,P)TF - r_0 T\mathbf{1}_{(\Xi,\Theta)},$$
$$\partial_t F = g_F(M,P)(G+S+T)F - b\Phi F - m_F(M,P,\pi)F - r(\delta)F\mathbf{1}_\Theta + \sigma(G+S+T)\xi_F$$
$$\partial_t B = m_G(P)G - g_G(P)BG,$$
$$\partial_t G = -\partial_t S - \partial_t T - \partial_t F - \partial_t B,$$
$$\partial_t \Phi = f(M,P,\pi,G+S) - \Phi + D\nabla^2 f(M,P,\pi,G+S).$$
$$(4)$$

The model has as external variables the spatial distribution of observed MAR ($P$), MSI ($M$), soil suitability for forest ($\pi$) and of the distance to the agricultural/urban class ($\delta$). These external variables influence growth and mortality rates ($g_i$ and $m_i$) in the equations. The $f$ is a stochastic variable that models local fire occurrence, with fire probability an increasing sigmoidal function of total fire-prone cover ($G+S$) and the cover value where this function increases steeply being a function of $M$, $P$ and $\pi$ (see Supplementary Fig. 4c,d). Fire diffusion between cells is captured by the term $D\nabla^2 f$ so that forest close to savanna will be more affected by fire than forest encircled by other forest. When a cell burns, a fraction $b$ of forest cover is removed. The $q$ represents sapling recruitment into adults and is a linearly decreasing function of fire. The choice of $g_i(M,P)$ is based on the assumption that growth rate saturates when water limitation is less severe. $m_i(M,P,\pi)$ was chosen to parameterize the increased mortality due to severe water limitation beyond a certain threshold. The only assumed human effect on savanna saplings and adult trees is a constant high removal rate $r_0$ in human-managed zones at times with human impact, or $(\mathbf{x},t) \in (\Xi,\Theta)$, where $(\Xi,\Theta)$ is the set of all locations and times with occurrence of human impact. Forest cover removal $r(\delta)$ is higher at closer distances $\delta$ to agricultural/urban areas at times when human impact occurs ($t \in \Theta$). We estimated soil suitability for forest $\pi$ is a variable constructed as follows,

$$\pi = \theta_1\varphi_s + \theta_2\varphi_c + \theta_3\rho + \theta_4\varphi_c\rho - \pi_c,$$
$$(5)$$

where parameters $\theta_i$ have been taken from the logistic regression model (Table 1) and $\pi_c$ is the value where the modelled effect of soil is zero. Finally, $\sigma\xi_i$ is spatially/temporally uncorrelated Gaussian noise with variance $\sigma^2$. The variance is further scaled with the relative availability of space that the cover type can colonize.

All plant species are conceptualized as cover fractions, such that for all time and space,

$$B + G + S + T + F = 1.$$
$$(6)$$

In a similar way to ref. 15, we have assumed the following competitive hierarchy

$$B < G < S < T < F$$
$$(7)$$

by allowing $F$ to colonize areas occupied by $S$, $T$ and $G$; $S$ to colonize areas occupied by only $G$; and $G$ to colonize areas only occupied by $B$. $G$ is also the default cover type, meaning that mortality of trees other than competition by other tree types

results in conversion to $G$. Bare soils can only form in dry places of grass cover. We assumed a continuous transition between bare soil and grass cover (and hence no grass–bare bistability). As shown by Supplementary Fig. 4a, savanna trees were assumed more drought resistant retaining low mortality and high growth rate for drier conditions than forest trees. The model functions and parameters used are shown in Supplementary Tables 2 and 3. The functions $g_i(M,P)$, $m_i(M,P,\pi)$, $f(M,P,\pi,G+S)$ are plotted in Supplementary Fig. 4a.

The fire component has the following form,

$$\partial_t\Phi = f(M,P,\pi,C) - \Phi + D\nabla^2 f(M,P,\pi,C),$$
$$(8)$$

where $C = G+S = 1-T-F-B$ represents fire-prone cover. Fire occurrence consists of local fire occurrence $f(M,P,\pi,C)$ and fire spread $D\nabla^2 f$. The chosen local fire occurrence function has a sigmoidal shape, with zero burning at low fire-prone cover, a steep increase around a critical cover value $C_c$ and maximal burning at cover value $C \gg C_c$ (Supplementary Fig. 4c,d), where $C_c$ is a function of $M$, $P$ and $\pi$ (Supplementary Fig. 4e,f). The $f(M,P,\pi,C)$ is shorthand for the random variable constructed via a Bernoulli trial Bernoulli ($p$) with

$$p = p(M,P,\pi,C) = \frac{1}{I}\frac{1}{1+\exp\{-k_f[C-C_c(M,P,\pi)]\}},$$
$$(9)$$

where $I$ is the minimum fire return interval, $C_c$ the fire-prone cover value where fire occurrence steeply decreases with $C$ and $k_f$ the steepness of this decrease. This logistic function is shown in Supplementary Fig. 4c,d as a function of $C$. Such a sigmoidal shape is similar to the ones derived from percolation models[16]. In percolation models, the percolation threshold is fixed. Here, we let this threshold vary as a function of climate and soils by choosing $C_c$ as a function of $M$, $P$ and $\pi$,

$$C_c(M,P,\pi) = 1 - \frac{1-C_0}{1+\exp[-k_M(M-M_0)+k_P(P-P_0)+k_\pi(\pi-\pi_0)]}.$$
$$(10)$$

A varying percolation threshold is not strictly necessary to get bistability in the model but it improves the comparability with fire data. Having a percolation threshold that is not fixed is a representation of the dependence of fire fuel effective connectivity and density with climate and soils.

In agreement with empirical studies, fire in our model affects savanna trees by reducing the recruitment of savanna saplings into savanna trees. We did this by choosing

$$q(\Phi) = q_0(1 - d\Phi)$$
$$(11)$$

such that recruitment is reduced with a factor $d$ in fully burnt areas. For a more detailed justification of the model structure with regard to empirical studies, see ref. 15. Mortality of forest trees by fire is affected by removing a fraction $b$ in burnt areas.

Near the forest edge, more area is affected by logging[26]. We assume an exponential decay of logging rate with distance from the forest edge,

$$r(\delta) = r_0 \exp(-k_r\delta)$$
$$(12)$$

In places where there is agriculture (where $(\mathbf{1}_{x\in\Xi})_i = 1$), clearcutting is modelled as a constant high removal rate ($r_0$) of trees and saplings caused by land management. In the transition zones, deforestation rate decays as a function of distance $r(\delta)$ from these agricultural areas. We further neglect diffusion of cover types ($D_F\nabla^2 F = D_S\nabla^2 S = 0$) compared with the effect of fire diffusion. Note hence that the only spatial interaction in our model comes from the fire spread term.

We use no-flux boundary conditions. The model is initialized ($t = 0$) with random cover values and run for 300 time steps, before which steady state was reached. Then, deforestation was switched on until the end of the simulation ($t = 500$), before which steady state was reached again.

**Discretized fire model and its interpretation.** After discretizing and showing spatial and temporal dependence in the fire equation, we obtain,

$$\Phi(t+1) - \Phi(t) = \mathbf{f}[\mathbf{M},\mathbf{P},\boldsymbol{\pi},\mathbf{C}(t)] - \Phi(t) - D\mathbf{Lf}[\mathbf{M},\mathbf{P},\boldsymbol{\pi},\mathbf{C}(t)],$$
$$\Phi(t+1) = (\mathbf{I} - D\mathbf{L})\mathbf{f}[\mathbf{M},\mathbf{P},\boldsymbol{\pi},\mathbf{C}(t)],$$
$$(13)$$

where we have written the spatial dependence as column vectors containing all the values at all discrete spatial locations. Using $\mathbf{L} = \mathbf{D} - \mathbf{A}$ (Laplacian matrix = degree matrix − adjacency matrix) and writing for one particular location,

$$\Phi_i(t+1) = f[M_i,P_i,\pi_i,C_i(t)] + D\sum_{\mathbf{x}_j\in N(\mathbf{x}_i)}\{a_{ij}f[M_j,P_j,\pi_j,C_j(t)] - f[M_i,P_i,\pi_i,C_i(t)]\},$$
$$(14)$$

where for every variable $X$, $X_i$ is shorthand for $X(\mathbf{x}_i)$ and $N(\mathbf{x}_i)$ refers to the neighbourhood of $\mathbf{x}_i$. Hence, fire occurrence at any location is then local fire occurrence plus $D$ times the differences with its nearest neighbours. We keep $\Phi < 1$ throughout the simulation.

The reason for choosing this fire model form is that fire percolates through space such that fires occurring on separate discretized spatial units cannot be considered as isolated from each other, and that several empirical studies have found that contagion of fires from cultivated/cleared areas into the forest are major causes of extra forest loss next to direct deforestation/logging[26–29]. Since areas close to the forest edge are often logged, this contagion process is reinforced. In our model, the contagion effect is modelled by the $D\nabla^2 f$ term, with $D$ the strength of this effect ($D = 0.1$ in our model).

**Phase diagram.** To delineate the boundary between savanna and forest in the diagram, the model was run on a $100 \times 100$ square lattice for every combination of values in a $100 \times 100$ $(\delta, P)$-parameter grid, storing the proportion of the lattice that is colonized by forest after 300 and 500 time steps, where forest means $F + T > 40\%$, starting from random initial conditions (with $S(0) + T(0) + F(0) + B(0) = 1$), and with periodic boundary conditions. Then, the difference in forest fraction was calculated by subtracting the earlier from the later simulated forest fraction. The MP occurs where this difference is zero and where the steady state forest fraction is not 0 or 1. Around the MP, convergence is very slow due to a slow front propagation speed. The treeless state was delineated based on the local model by finding for which value of $P$ savanna tree cover becomes zero. By fitting an exponential function to the MP in $(\delta, P)$, the forest state could be predicted in space (Fig. 6b), with the effect of $M$ and $\pi$ taken into account by substituting in this exponential equation $P$ by

$$P - \frac{K_M}{K_P}(M - \bar{M}) + \frac{K_S}{K_P}(\pi - \bar{\pi} + \pi_0). \tag{15}$$

**Software.** *Data.* All geoprocessing was done with an interface of GRASS7 (grass.osgeo.org) with SciPy (scipy.org), using the GRASS-Python scripting library (grasswiki.osgeo.org/wiki/GRASS_Python_Scripting_Library).

*Modelling.* We used MATLAB 2015a (uk.mathworks.com) to run the model using our own vector Euler–Maruyama scheme. We used time step $\Delta t = 1y$ and spatial resolution $\Delta x = 1\,\mathrm{km}$. Such a high resolution was necessary to model fire contagion, human impact and their interaction that occur on small spatial scales. The required memory was provided by the high memory nodes of Blue Crystal, the high-performance computer of the University of Bristol (bris.ac.uk/acrc).

*Plots and figures.* Plots and figures were created using SciPy, Matlab 2015a and Mathematica 10 (wolfram.com/mathematica). Maps were drawn with GRASS7.

**Data availability.** All data used in this study except the fire occurrence data are publicly available from the web. The original forest cover dataset is available from the UNEP-WCMC database (old.unep-wcmc.org/generalised-original-and-current-forests1998_718.html). VCFs are available from the Land Processes Distributed Active Archive Center (lpdaac.usgs.gov/dataset_discovery/modis/modis_products_table/mod44b). The TRMM 3B43 rainfall data set is available from the Tropical Rainfall Measuring Mission database (trmm.gsfc.nasa.gov). Soil information is available on the Harmonized World Soil Database (webarchive.iiasa.ac.at/Research/LUC/External-World-soil-database/HTML). Crop and pasture intensity information can be found on Earth Stat (www.earthstat.org). The MODIS IGBP Land cover product is available from the Land Processes Distributed Active Archive Center (lpdaac.usgs.gov/dataset_discovery/modis/mod-is_products_table/mcd12q1). The SRTM digital elevation model can be found on CGIAR-CSI database (srtm.csi.cgiar.org). WWF Terrestrial ecoregions can be found on the World Wildlife Fund website (www.worldwildlife.org/publications/terrestrial-ecoregions-of-the-world). The computer code used for this study is available from the corresponding author on request.

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

## Acknowledgements

We thank Rich Pancost, Milena Holmgren, Arie Staal, Simon Gilson, Delphine Zemp, Sourav Das, Gert Verstraeten and Gerard Govers for feedback and discussion. We also thank Renata Libonati for sharing the burnt area data. For our simulations, we made use of Blue Crystal, the supercomputer of the Advanced Computing Research Centre, University of Bristol. Our work was funded by UK Engineering and Physical Sciences Research Council (to B.W. and A.R.C.) and European Union LUC4C project (to J.I.H.).

## Author contributions

B.W. performed the data analysis and constructed the model under supervision of A.R.C. and J.I.H. B.W., A.R.C. and J.I.H. wrote the paper.

## Additional information

**Competing interests:** The authors declare no competing financial interests.

**DOI: 10.1038/ncomms16179**   **OPEN**

# Author Correction: Amazonian forest-savanna bistability and human impact

Bert Wuyts, Alan R. Champneys & Joanna I. House

*Nature Communications* 8:15519 doi:10.1038/ncomms15519 (2017); Published online 30 May 2017; Updated 21 Feb 2018

In the originally published version of this Article, reference 23 did not refer to the correct paper. This error has now been corrected in both the HTML and PDF versions of the Article.

