## [Peer Review File · Nature Communications]

Reviewers' Comments:

Reviewer #1 (Remarks to the Author)

A. Summary

This paper develops the idea that forest-savanna bistability is enhanced in areas with more human impact, and less extensive (they argue absent) in areas remote (e.g. > 3 km) from urban areas, agriculture and pasture. In these 'human influenced' zones they suggest that selective (or small scale) logging allows fire to creep into the forest and blur the forest-savanna boundary. The most interesting point is that hysteresis in systems with inherent bistability can be reduced or removed by spatial interactions. That is, our conceptual and numerical models for forest-savanna transitions routinely bifurcate because of stochastic fire when used in 'point-mode' (including the model these authors use). When spatial interactions are included, however, that tendency to bifurcation is buffered and a more continuous (Maxwell Point) emerges in the spatial model.

The authors have revised their previous submission to Nature journals. The new version is rather more complex, with a new index of forest suitability derived from an environmental model fit to data on "potential forest cover". While interesting, this addition makes the paper much more complicated and the authors expand the text to discuss forest distributions based on the potential cover which distracted a little from the primary message that spatial interactions may reduce hysteresis in systems prone to bifurcation.

B. Main Points

1. The analysis is original and significant, in alerting readers interested in alternate stable states in savanna-forest (and other systems) to the importance of spatial interactions. A complimentary paper by Staal et al. (Ecosystems (2016) 19: 1080–1091) investigates similar processes, but the approach in this manuscript is distinct and the theme is certainly important enough to merit treatment in multiple papers.

2. I think this paper is a significant contribution, but I would encourage the authors to focus on the main theme of spatial interactions buffering bifurcations and reducing hysteresis. They could, for example, return to analyzing tree cover variation against mean annual rainfall (Fig S3 seems to justify it!) or using a drought index (per previous versions) to keep it simple and avoid confusing the reader with a new index (CEFS).

3. CEFS is based on a model of potential forest distribution. If they retain the WCMC and CEFS approach, I would suggest reducing text on the distribution of "potential" cover. Is there a danger of circularity in modeling the results of an earlier model? Does the fitted model of potential forest distribution provide new insight into the environmental drivers, beyond the insight which presumably went into creation of the WCMC?

4. Figure 3: given quite significant differences between observed tree cover patterns and the model in Figure 3 (the full model in row B is much less continuous (bifurcates more than) the observations in A), are you confident the model is good enough to be used to generate Figure 4?

C. Minor Points

Line 72: "results" is not defined

Line 127: this sentence is very hard to understand!

Supplement Line 37: See also and reference Hanan et al. *Global Ecology and Biogeography*, (*Global Ecol. Biogeogr.*) (2015) 24, 988–989

Figure S3: I found this plot to be one of the most interesting in the manuscript: showing the clear MAP relationship and importance of dry season length in driving the “bifurcation” among high and low tree cover locations. This might be a fire feedback, but could of course also reflect dry season seedling mortality. I would emphasize this plot by bringing it into the main text.

Niall Hanan

Reviewer #2 (Remarks to the Author)

Wuyts et al. present an interesting analysis on contrasting theories about the dynamics of tropical savanna – forest boundaries. Previous studies have suggested that at intermediate precipitation levels forest and savannas may exist as two alternative stable states under similar climatic conditions. The implications of the suggested hysteresis are large and important for understanding ecosystem stability in the future.

The mechanisms that may drive such alternative stable states have been observed on small scales, and many savannas contain pockets of forest. Forests are generally better able to maintain moist conditions during the dry season and trees reduce herbaceous biomass production, both reducing fire return intervals and fire intensity. However, the extent to which these processes play out on larger scales is more difficult to resolve. Several studies have suggested that the spatial distribution of tree cover as observed by satellite provides evidence for large scale alternative stable states and the occurrence of hysteresis.

Here, the authors develop a model to show that a sharp forest savanna boundary can also explain the spatial distribution of tree cover in South America, if logging and fire are included. Following Good et al. (2016) they find that the occurrence of alternative stable states is not needed to reproduce spatial patterns of tree cover as observed by satellite. The authors present a compelling alternative for the “alternative stable state” theory, and the work is suitable for publication in nature communications, after addressing my suggestions.

Major comment

The authors demonstrate that using a relatively simplistic model, based on climate, soil and human influences, they can reproduce the savanna forest boundary and a similar distribution of tree cover as observed by satellite for tropical South America. Given the pervasive influence of humans on tree cover in most of the South American savannas and the deforestation areas they show that their model is in addition able to capture historic distribution of tree cover, although some uncertainty remains at the forest edges. Their model suggests, no hysteresis is needed to reproduce observed patterns. Moreover, the authors demonstrate that much of the evidence for bistability is in fact caused by human patterns of land cover conversion and their influence on tree cover distribution. Therefore, there is little proof for hysteresis, or the existence of large “bistable zones”, suggested by some previous studies.

Yet, the fact that their model produces such a sharp line between forest and savannas may not be entirely surprising, given their model design. Many of the processes that could cause a mixed landscape of forest and savanna patches are not represented within their model (e.g., a vegetation climate-feedback, heterogeneity of the landscape (beyond general soil types), and the influence of climate variability). The fact that their model does not incorporate vegetation – climate interaction, may be the reason they are not seeing two alternative stable states on a smaller scale, closer to the MP. Even in the natural vegetation (Fig. 3Ac) there is a wide range of possible tree cover fractions observed at -5 to 2.5 forest suitability, that their model does not capture (Fig. 3Bc).

The authors do not convincingly rule out the possibility of a hysteresis effect on a smaller scale, that could result in a broader transition zone around the MP. It would help the manuscript if they would discuss this concept/the possibility of such a transition zone. This would not change the importance of the MP since the probability of encountering a "stable forest" on the savanna side of the MP would still decline with distance and vice versa. But it would help to understand how their theory is in line with the observed patchy landscape in many places (beyond the influence of humans alone).

Minor comments

It would be good to include recent studies by Good et al. (2016) and Oliveras et al. (2016).

Fig. S4: in the caption, "D-F" should be "C-E"

Fig. S6: "A" is missing for subplot S6A.

References

Good, P., Harper, A., Meesters, A., Robertson, E. and Betts, R.: Are strong fire-vegetation feedbacks needed to explain the spatial distribution of tropical tree cover?, *Glob. Ecol. Biogeogr.*, 25, 16–25, 2016.

Oliveras, I. and Malhi, Y.: Many shades of green: The dynamic tropical forest-savanna transitions, *Philos. Trans. Ser. B, Biol. Sci.*, 371: 20150, 2016.

Reviewer #3 (Remarks to the Author)

I have read this manuscript by Wuyts et al. with great interest and believe it is of considerable merit. The manuscript goes beyond previous work on a widely relevant topic in both empirical and theoretical ways. It thus moves forward our understanding of the drivers of South American tree cover, and the modeling efforts will open up new possibilities for analyzing the behavior of this system. Although these results therefore deserve to be published, I have concerns regarding the interpretation of those results and the conclusions drawn from them. To quote Carl Sagan, "Extraordinary claims require extraordinary evidence", and given that this manuscript draws firm conclusions that oppose a growing body of literature on forest-savanna bistability, the question arises how extraordinary the evidence really is.

The authors' main conclusion is that no hysteresis exists between Amazonian (rather, tropical South American) forests and savannas under circumstances free from human influence. In this regard they contradict the conclusions of previous studies that, using remote sensing data, showed bimodalities of tree cover for a range of climatic conditions in support of the hysteresis hypothesis (1, 2). If the environmental overlap of the high and low tree cover modes indeed do not exist, this would have major implications for predictions on the effects that environmental changes may have on the transitions between Amazonian forests and savannas. The authors conclude that by considering additional factors like soil texture, and assuming the influence of humans to extend further into forests than previous studies did, the distribution of high and low tree cover in the Amazon becomes predictable from environmental variables alone.

I am not convinced of this environmental determinism of forest that the authors claim to find, for a number of reasons:

- As said, previous work has demonstrated bimodalities in tree cover in South America under certain environmental conditions, indicating forest-savanna hysteresis. The crucial point in this

manuscript is that such bimodality does not exist for any climatic-edaphic conditions >3km away from human-influenced areas. However, when I look at Fig. 2D I see bimodality of high and low tree cover over a range of ~2 units of CEFS. The fact that this range is less than half the length of that when transition zones are included (Fig. 2C) indeed indicates that much of the observed bimodalities occur in the transition zones. However, one might also argue that given that—even after accounting for average rainfall, seasonality of rainfall, soil sand content, soil clay content, soil bulk density, wetland vs. non wetland, dry forest vs. non dry forest, montane forest vs. non-montane forest, and adopting a conservative approach in accounting for human effects—the bimodalities still have not disappeared, this manuscript actually supports (but nuances), rather than contradicts, the hypothesis of Amazonian forest-savanna hysteresis.

- The data selection has been very strict, especially by removing dry forests (by definition forests found in climates out of the ordinary) and wetlands, so a full comparison with (1, 2) is not possible.

- “Edge effects” on forest fragments (e.g. ref. 3), like microclimatic changes, impose a negative effect on tree cover without direct human interference like logging. By assuming that human influence extends up to 3 km beyond agricultural areas, non-logging- and fire-related negative effects on tree cover near forest edges could have been excluded.

- Based on what has the transition zone been defined as the area within 3 km from human-used areas? I could not find an a priori justification, suggesting that it has been selected based on some goodness-of-fit. But this is concerning since, judging by Fig. 2B, very many areas have been excluded, in particular in the cerrado.

Some more minor remarks:

- Low tree cover areas near human-affected areas (the transition zones) are not considered true savannas but rather degraded forests. The authors mention in their reply to referee 1 in the previous round that there exist no large-scale observations about the occurrence of savanna vs. forest trees. However, savanna trees are lower than forest trees, and a recent study (4) related tree cover to remotely sensed canopy height; they found large correspondence between the high tree cover mode and a high canopy height mode, characteristic of forests (and between a low cover mode and low height mode, characteristic of savannas). Nevertheless, they also found that some low cover areas had high canopies, indicating degraded forests. It is plausible that these degraded forests occur relatively often in the human-natural transition zones as defined here (note that this cannot be checked with the coarse-resolution canopy height data), but at least those results are an indication that low tree cover generally rightly captures savanna tree species.

- In lines 72-73 it is mentioned that this work studies the impact of humans on Amazonian forest-savanna bistability. However, the question how humans affect this bistability is not answered explicitly. It is sufficiently clearly articulated that there exists no bistability without human impact, but do the authors believe that there is bistability with human impact? Or is it just apparently so, because low tree cover in the transition zones is artificial?

- It is not without reason that some previous studies did not account for soil variables in their analyses: the quality of the soil databases does not match that of tree cover, so the matter of soil effects cannot be settled in a conclusive way yet. Using available data is definitely better than nothing, but this limitation should be acknowledged in the manuscript.

- Lines 259-261: forests might grow more easily in a degraded forest area than in savanna, so a degraded forest does not need to be an alternative stable state to a non-degraded forest; but that does not mean that forest and savanna would not be alternative stable states.

In summary: although I am impressed by the empirical and theoretical work (and their connection) being done here, I believe that the conclusions are not as straightforward as the authors write. This means that this manuscript does not settle the discussions in the literature about whether Amazonian forests and savannas can exhibit hysteresis. If the authors decide to keep the analyses as they are, I believe that the limitations above should be acknowledged. If they decide to re-do analyses without doing some of the criticized assumptions, it may be that some of the conclusions of the paper would need to be changed.

Literature

1. Hirota M, Holmgren M, Van Nes EH, & Scheffer M (2011) Global resilience of tropical forest and savanna to critical transitions. *Science* 334(6053):232-235.
2. Staver AC, Archibald S, & Levin SA (2011) The global extent and determinants of savanna and forest as alternative biome states. *Science* 334(6053):230-232.
3. Laurance WF, Ferreira LV, Rankin-de Merona JM, & Laurance SG (1998) Rain forest fragmentation and the dynamics of Amazonian tree communities. *Ecology* 79(6):2032-2040.
4. Xu C, et al. (2016) Remotely sensed canopy height reveals three pantropical ecosystem states. *Ecology* 97(9):2518-2521.

With regard to Reviewer 2 and 3's concerns about bimodality and hysteresis and the extent to which our results contradict previous hypotheses, we should like to set out the following.

General reply to main criticism of Reviewers 2 and 3

- We accept that there was some confusion in the previous version about what we mean by the terms **bimodality**, **bistability** and **hysteresis**. In particular bistability and hysteresis are system-dynamic properties while bimodality only refers to the statistical distribution of a variable.
- Specifically we now consistently use **bimodality** to refer to overlapping forest and savanna states in scatterplots of tree cover vs. rainfall (or other relevant input variables). We used the concept of hysteresis to explain the connection
- Our results show there is little bimodality left after taking into account the joint effect of average annual rainfall, rainfall seasonality and soils or CEFS, but excluding zones that are human impacted.
- This observation is consistent with the theory of spatially extended ecosystems, where spatial interaction in bistable systems causes the system to topple into the most stable state. The Maxwell point (MP) is the value of external parameters (rainfall, soil etc) at which the most stable state undergoes a transition between the two states (savanna and forest in this case).
- Note that this does not mean that we oppose the alternative stable states hypothesis. In the model without diffusion, forest and savanna are indeed alternative stable states, but with diffusion, only the most stable one is observed. Only around the MP, would both be equally preferred, that is true **bistability**. Hence, while previous studies inferred just from rainfall that there are large areas where both savanna and forest are possible, we show a more nuanced picture and infer from rainfall and other relevant variables that this area is restricted to the contour in 2D space (of CEFS and distance from human impact) where the MP occurs.
- The system we are studying is indeed **locally bistable** in that its local dynamics (=without spatial interaction) has two alternative stable states for a certain value range of the input parameters. This would lead to **hysteresis** — that is, different transition points upon increase of a parameter than

under decrease of that parameter. When there is spatial interaction in the model though, bistability only occurs at the Maxwell point. Hence, whereas the local system (without diffusion) is bistable, there is no hysteresis.

- To avoid confusion, we distinguish in the text between local and global bistability. Local bistability is the occurrence of alternative stable states for a specific range of parameter values in the system without spatial interaction. We talk about **global bistability** if a system has alternative stable states despite existence of spatial interaction. Hence, while our model is locally bistable, it is not globally bistable.
- Nevertheless, we do accept that there is some remaining bimodality in the data after dealing with human impact and natural spatial heterogeneity. This can indeed be explained by a smaller-scale hysteresis that we have not modelled, or could equally well be explained by further spatial heterogeneity of the Maxwell Point (MP) as a consequence of its dependence on other variables than the ones we considered in our work. If there is hysteresis on a smaller scale, the reviewers rightly point out that based on the data, the MP is rather a range between contours than a single contour. We have added text and adjusted the discussion about this in the main article. and have tried our best to say that our results are not a direct contradiction to previous hypotheses but point out that the true situation is more nuanced.

Detailed reply to Reviewer 1

Main Points

The authors have revised their previous submission to Nature journals. The new version is rather more complex, with a new index of forest suitability derived from an environmental model fit to data on “potential forest cover”. While interesting, this addition makes the paper much more complicated and the authors expand the text to discuss forest distributions based on the potential cover which distracted a little from the primary message that spatial interactions may reduce hysteresis in systems prone to bifurcation. I think this paper is a significant contribution, but I would encourage the authors to focus on the main theme of spatial interactions buffering bifurcations and reducing hysteresis. They could, for example, return to analyzing tree cover variation against mean annual rainfall (Fig S3 seems to justify it!) or using a drought index (per previous versions) to keep it simple and avoid confusing the reader with a new index (CEFS). CEFS is based on a model of potential forest distribution. If they retain the WCMC and CEFS approach, I would suggest reducing text on the distribution of “potential” cover.

- That the use of the new index caused distraction indicates that we had not succeeded sufficiently in explaining its superiority compared to other indices and its relevance to the primary message. We have revised the paper to better explain how plotting tree cover as a function of CEFS minimises the confounding effect of natural spatial heterogeneity. The idea is that if you plot tree cover as a function of MAP, the independent variation of MSI or soils can cause variation in the tipping points that is not due to hysteresis. We dropped MAWD because although MAWD contains information about both rainfall seasonality and rainfall averages (Supplementary Figure 8), there is no reason to assume that MAWD is an optimal synthesis of these two predictor variables. Also, MAWD does not take into account the edaphic effects. Therefore, remaining overlap in the scatterplots of natural tree cover versus MAWD could still be due to independent effects of rainfall averages or seasonality, or due to soils. The use of CEFS eliminates this possibility as much as possible. By knowing that CEFS predicts natural forest cover well we can state with confidence that bimodality over wide ranges of CEFS indicates other than natural influences. Furthermore, the main contributions to CEFS, which are MAP and MSI are linearly combined. This makes the role of rainfall averages and of seasonality — the two most important effects cited in the literature — much more clear in CEFS than in MAWD.
- We have although, as the reviewer suggests, tried to emphasise our main message better throughout the paper.

Is there a danger of circularity in modeling the results of an earlier model?

- The WCMC map does not only rely on previous potential forest cover maps but also on satellite data and expert knowledge on ecoregions. When comparing the pre-human forest data to the LANDSAT-based PRODES original forest data in the Brazilian Amazon, the accuracy of the pre-human forest data is high. For the Brazilian Amazon, the pre-human forest cover data agrees well with the pre-1988 forest cover map based on LANDSAT, with small deviations near some of the edges (Figure 1, this document).

Does the fitted model of potential forest distribution provide new insight into the environmental drivers, beyond the insight which presumably went into creation of the WCMC?

- The necessary insight for our study is that if the all of the most relevant predictors for forest are used together, forest can be predicted much better than based on only one predictor. Even if that insight was used to create the WCMC map, it was not recognised sufficiently in previous studies about forest-savanna bistability.

Figure 3: given quite significant differences between observed tree cover patterns and the model in Figure 3 (the full model in row B is much less continuous (bifurcates more than) the observations in A), are you confident the model is good enough to be used to generate Figure 4?

- The somewhat more continuous transition in the data in the range $CEFS = [-5, 0]$ is we believe due to stochasticity or heterogeneity that are not captured by the model. The fact that this is in the zone where savanna occurs suggests that this has to do with processes that are relevant in savannas, especially fire, drought and their interaction.

Figure 1: Map of pre-human forest cover (transparent blue) overlaid on a map of 2010 PRODES deforestation data (green: forest in 2010, yellow: deforested between 1988 and 2010, pink: no forest, light blue: cloud).

Although in the data some patches have attained higher tree cover values, the majority of them have experienced large decreases in tree cover (large negative median of tree cover change in Figure 4B). Effects that might cause such effects include the following:

- **Stochasticity in aspects of fire/drought.** The used burnt area model does not model the size distribution of the burnt patches. A model that does take into account the size distribution would produce a spatial distribution of burnt area that has more clustering, with some large fires and many small (assuming a power-law like distribution of burnt area). If burnt area were more clustered, the total perimeter of pixels with fire in the spatial domain would be smaller. Therefore, there would be more pixels that are not neighbouring burnt pixels and hence less diffusion of burnt area. Consequently, there is a higher chance that in some locations, tree cover will regrow more between fires. We expect that inter-annual variability reinforces this stochasticity by adding year to year variation in the location of fire-prone areas.
 - **Heterogeneity in aspects of fire/drought resistance of savanna saplings/trees.** If there exist sub-populations of savanna saplings/trees that are less affected by drought or fire, they will attain higher tree cover values. This concerns the model parameters q_0 (sapling recruitment), d (reduction of sapling recruitment by fire) and climate-related growth/mortality rate functions of savanna saplings and trees.
 - **Heterogeneity of in aspects of fire/drought resistance of forest trees.** Rather than a discrete classification of dry versus moist forest, it may be that there is a range between them as a consequence of existence of several adaptation strategies or phenotypic plasticity.
- These effects would be interesting to study in future work, but the failure to capture these precise details would hardly seem to detract from the otherwise good agreement between our model output and the data.

Minor Points

Line 72: "results" is not defined

- We have now replaced 'results' with 'forest stability'.

Line 127: this sentence is very hard to understand!

- We have replaced it with a better explanation now.

SI Line 37: See also and reference Hanan et al. Global Ecology and Biogeography, (Global Ecol. Biogeogr.) (2015) 24, 988–989

- OK. Thank you.

SI Supplementary Figure 3: I found this plot to be one of the most interesting in the manuscript: showing the clear MAP relationship and importance of dry season length in driving the “bifurcation” among high and low tree cover locations. This might be a fire feedback, but could of course also reflect dry season seedling mortality. I would emphasize this plot by bringing it into the main text.

- Thank you for the suggestion. We have brought an equivalent figure in the main document now (Figure 3) and explained its context.

Reply to Reviewer 2

Here, the authors develop a model to show that a sharp forest savanna boundary can also explain the spatial distribution of tree cover in South America, if logging and fire are included. Following Good et al. (2016) they find that the occurrence of alternative stable states is not needed to reproduce spatial patterns of tree cover as observed by satellite. The authors present a compelling alternative for the “alternative stable state” theory, and the work is suitable for publication in nature communications, after addressing my suggestions.

- Thank you for mentioning this new study. We do want to clarify though that we do not offer an alternative but only present a more subtle picture. Good et al. argue that fire-vegetation feedbacks are not necessary, whereas we included them in our work. Good et al. are indeed correct that alternative stable states can be produced when only assuming influences of climate. However, there is evidence from independent lines of research for existence of fire-vegetation feedbacks (e.g. Pausas et al., in press). Local-scale forest distribution affects fire occurrence, and as forest is also affected by fire, this generates a feedback between fire and forest. Figure 2 in this document shows a sub-region in the study area with burnt area between 2005 and 2014 (red) and forest occurrence of 2005 (tree cover >40%, dark green, burnt forests indicated in blue). It shows that fire occurs almost exclusively in areas of low tree cover, on a much finer scale than what can be expected based on just a climatic influence on fire. The fire feedback that we assumed has two terms: (i) a local sub-pixel term, which is a parameterisation of the fire feedback consistent with fire percolation models (sigmoidal-shaped function), and (ii) a spatial interaction term, accounting for spilling into neighbouring pixels (diffusion). The local term varies in space with climate and with fire-prone cover. Our model results are, as Good et al.’s model results, consistent with the intermediate fire-productivity hypothesis (see Supplementary Figure 5B).

Pausas J.G. & Dantas V.L. (in press). Scale matters: Fire-vegetation feedbacks are needed to explain tropical tree cover at the local scale. *Global Ecology and Biogeography*

Major comment

Yet, the fact that their model produces such a sharp line between forest and savannas may not be entirely surprising, given their model design. Many of the processes that could cause a mixed landscape of forest and savanna patches are not represented within their model (e.g., a vegetation climate-feedback, heterogeneity of the landscape (beyond general soil types), and the influence of climate variability). The fact that their model does not incorporate vegetation – climate interaction, may be the reason they are not seeing two alternative stable states on a smaller scale, closer to the MP. Even in the natural vegetation (Fig. 3Ac) there is a wide range of possible tree cover fractions observed at -5 to 2.5 forest suitability, that their model does not capture (Fig. 3Bc). The authors do not convincingly rule out the possibility of a hysteresis effect on a smaller scale, that could result in a broader transition zone around the MP. It would help the manuscript if they would discuss this concept/the possibility of such a transition zone. This would not change the importance of the MP since the probability of encountering a “stable forest” on the savanna side of the MP would still decline with distance and vice versa. But it would help to understand how their theory is in line with the observed patchy landscape in many places (beyond the influence of humans alone).

- Indeed, we did not include all processes. We do believe however that we have captured the most relevant ones, i.e. spatial heterogeneity due to climate and soil. We agree with the referee that the remaining overlap still exists due to processes/phenomena that are not captured by the model (if we ignore data inaccuracies) and we have indeed adjusted the Discussion of the paper accordingly. Specifically these effects are:
 - **Vegetation-climate feedback.** Earlier studies based on simple non-spatial models have shown that vegetation-climate feedbacks can affect the derived region of bimodality as inferred from data (ref. 13) and that feedbacks on smaller scales can interact with feedbacks on larger scales (ref. 46). A model with such a feedback was found to exhibit larger hysteresis than one without this feedback (ref. 13). Although the Van Nes model (ref. 13) showed the importance of taking into account hydrological feedbacks, it is hard to believe that it can accurately represent

Figure 2: A sub-region in the study area with burnt area between 2005 and 2014 (red) and forest occurrence of 2005 (tree cover $>40\%$, dark green). Burnt areas in forests are indicated in blue. Masked areas (wetlands, dry forests, highlands/montane forests) are in grey. Note that the burnt areas almost exclusively occur outside forests. If they occur in forest, they are at the forest edges. Hence, while it is well known that fires affect forest, it is also the case that forests affect fires. This two-way interaction is enough to have feedbacks.

the effect of hydrological feedbacks on inferred bimodality in scatterplots of tree cover versus rainfall. The reason is that both the local fire-vegetation and the regional hydrological feedbacks exist due to spatial interactions, which are ignored in the Van Nes model. Also, no information about the scale of the feedbacks was taken into account. Another issue is that rainfall recycling is expected to occur at different intensities in different sub-regions of the study area. This would introduce extra heterogeneity. A logical next step would hence be to couple a spatio-temporal ecohydrological model to our model to see how it affects the results.

- **Climatic trends.** In the Van Nes study (ref. 13), it was found that besides gradual climatic drying/wetting led to an inferred hysteresis that is shifted towards higher/lower rainfall compared to the real one.
- **Fire size distribution effects.** Our fire model does not model fire explicitly. It just captures the nonlinear dependence of fire on tree cover, climate and soils, and the effect that neighbouring $1km^2$ pixels are influenced by each other. A more realistic fire model may start from a distribution of fire sizes conditional on tree cover, climate and soils, with climate not only a function of space but also of time. This more realistic model will generate some rare but large fires during droughts that may be responsible for a considerable part of the fire frequency observations. This would result in spatial burning patterns that are more contiguous, with incursions into the forest at places where climatic and soil suitability for forest increase less steeply. We expect this more realistic dynamics could well lead to a blurred Maxwell region rather than a crisp Maxwell point, due to their stochastic nature. As we did not take these effects into account in our model, we should indeed expect larger prediction errors close to the Maxwell point (front), which is indeed what we see.
- **Climate variability and extreme events.** Droughts lead to temporally increased fire occurrence and intensity, especially around impacted zones (refs 28,44). On the other hand, during wet extremes, extensive tree recruitment can overwhelm disturbance (ref. 45)
- **Landscape heterogeneity of cover types related to other phenomena than climate and soils,** such as topography, radiation, temperature, variation of plant physiology, fire/drought resistance of forest.
- **Natural edge effects beyond those of fire.** Edge effects involve increasing mortality closer to the forest edges. In our model, the edge effects of fire are taken into account by having it creeping into neighbouring areas. As this is exactly what causes the front to form, fire-related edge effects are unlikely to be a cause of the remaining overlap. However, edge effects related to microclimatic differences and increased windthrow (ref. 43) may explain some of the remaining overlap. As most of those effects occur within a couple of hundreds of meters from the edge (ref. 25), the overlap they cause is expected to be much smaller than that caused by logging.

Minor comments

It would be good to include recent studies by Good et al. (2016) and Oliveras et al. (2016).

- We have now added them. Thank you.

Fig. S4: in the caption, “D-F” should be “C-E”. Fig. S6: “A” is missing for subplot S6A.

- Thanks for spotting this. It has been corrected now.

References (by referee):

Good, P., Harper, A., Meesters, A., Robertson, E. and Betts, R.: Are strong fire-vegetation feedbacks needed to explain the spatial distribution of tropical tree cover?, *Glob. Ecol. Biogeogr.*, 25, 16–25, 2016.

Oliveras, I. and Malhi, Y.: Many shades of green: The dynamic tropical forest-savanna transitions, *Philos. Trans. Ser. B, Biol. Sci.*, 371: 20150, 2016.

Detailed reply to Reviewer 3

In summary: although I am impressed by the empirical and theoretical work (and their connection) being done here, I believe that the conclusions are not as straightforward as the authors write. This means that this manuscript does not settle the discussions in the literature about whether Amazonian forests and savannas can exhibit hysteresis. If the authors decide to keep the analyses as they are, I believe that the limitations above should be acknowledged. If they decide to re-do analyses without doing some of the criticized assumptions, it may be that some of the conclusions of the paper would need to be changed.

- We have chosen the first option.

Major comments

I have concerns regarding the interpretation of those results and the conclusions drawn from them. To quote Carl Sagan, “Extraordinary claims require extraordinary evidence”, and given that this manuscript draws firm conclusions that oppose a growing body of literature on forest-savanna bistability, the question arises how extraordinary the evidence really is.

- We did not set out to make extraordinary claims, but ones that are consistent with the theory of locally bistable systems under the influence of spatial diffusion. Rather than oppose the literature as such, we have tried to point out that the story is in fact more subtle. We have now rewritten the introduction and conclusion to try to tell the whole story more explicitly

The authors’ main conclusion is that no hysteresis exists between Amazonian (rather, tropical South American) forests and savannas under circumstances free from human influence. In this regard they contradict the conclusions of previous studies that, using remote sensing data, showed bimodalities of tree cover for a range of climatic conditions in support of the hysteresis hypothesis (1, 2).

- We did not argue that human impact introduces hysteresis. It just makes it appear as if there is. In general, it is the confounding effect of other variables that influence the probability of being in a particular (tree/savanna) state that leads to overlapping states in tree cover vs. rainfall plots, which had been interpreted previously as hysteresis (see section our reply to the Editor for a more detailed explanation).

I am not convinced of this environmental determinism of forest that the authors claim to find, for a number of reasons:

- We are not in a strict sense arguing for determinism as such, indeed, even our model contains stochastic terms. Rather, we have shown that in the absence of human influence, both the data and the model support a much sharper transition with rainfall (or CEFS) between forest and savanna than had been previously reported, with little evidence of global bistability. We have also argued that this is a natural consequence of dynamical systems in the presence of spatial interaction, due to the Maxwell point argument.

- As said, previous work has demonstrated bimodalities in tree cover in South America under certain environmental conditions, indicating forest-savanna hysteresis. The crucial point in this manuscript is that such bimodality does not exist for any climatic-edaphic conditions >3km away from human-influenced areas. However, when I look at Fig. 2D I see bimodality of high and low tree cover over a range of ~2 units of CEFS. The fact that this range is less than half the length of that when transition zones are included (Fig. 2C) indeed indicates that much of the observed bimodalities occur in the transition zones. However, one might also argue that given that—even after accounting for average rainfall, seasonality of rainfall, soil sand content, soil clay content, soil bulk density, wetland vs. non wetland, dry forest vs. non dry forest, montane forest vs. non-montane forest, and adopting a conservative approach in accounting for human effects—the bimodalities still have not disappeared, this manuscript actually supports (but nuances), rather than contradicts, the hypothesis of Amazonian forest-savanna hysteresis.

- Indeed, there is some limited remaining overlap. This could be due to hysteresis, for example as a consequence of vegetation-climate feedback, climate variability or fire size distribution effects, but not necessarily. Spatial heterogeneity that is not captured by the model can also introduce variability in the location of the Maxwell point, leading to some overlap. We have discussed both of these viewpoints in the revised version of the text.

Figure 3: Scatterplots of tree cover versus CEFS with dry/montane forests included (A) compared to when they are excluded (B = Figure 4A-C). Greyed out places are combinations of CEFS and tree cover that have low frequencies or for (A) also places with erroneously high changes in tree cover. Note: to keep (A) comparable to previous studies, we have also not excluded areas with low quality values. These comprise some pixels in the Eastern Amazon erroneously registered in 2010 with low tree cover values due to cloud interference with satellite data. As this error does not occur as much in 2000, the calculated change between 2000 and 2010 was unrealistically high in these areas (50-60% in 10 years). Therefore, only changes up to +20% are shown in (A), with higher values greyed out. The grey in other sections of the figure indicate places where the sample size is small.

- *The data selection has been very strict, especially by removing dry forests (by definition forests found in climates out of the ordinary) and wetlands, so a full comparison with (1, 2) is not possible.*

- We have created a figure that is more comparable to previous studies by including the ecoregions of dry and montane forests. Excluding the ecoregions of dry and montane forests led to a noticeable decrease in the occurrence of forest (tree cover > 40%) between CEFS=-6 and CEFS=-2 (Figure 3 in this document). This is what can be expected as their adaptations to drought and fire lead to a Maxwell point at lower CEFS. Despite the increased noise when including dry/montane forests, the general pattern is still the same as when they are excluded. Due to their distinct adaptations, they should be treated as a separate cover type if modelled. They are, however, relatively limited in areal extent compared to moist forests and occur in separate regions. Therefore, their adaptations to drought and fire may be distinct for every region. Besides that not enough seems to be known about how and in which proportion dry forests are affected by drought and fire, we thought that adding an equation for dry forest would complicate the model too much compared to the possible added explanatory power. Wetlands are very limited in areal extent and do hence not influence the result considerably.

- *“Edge effects” on forest fragments (e.g. ref. 3), like microclimatic changes, impose a negative effect on tree cover without direct human interference like logging. By assuming that human influence extends up to 3 km beyond agricultural areas, non-logging- and fire-related negative effects on tree cover near forest edges could have been excluded.*

- Indeed, very often, the forest-savanna edge occurs within 3km from human impact. Nonetheless, remember that we performed stratified sampling. Hence, as long as there are some remaining natural forest-savanna edges that are reasonably representative for the whole study area, the resulting plots should be unbiased. But this is an important point to acknowledge. Also as fire-unrelated edge effects may be an additional cause of remaining overlap (see also 3.2.1). Thank you for mentioning it.

- *Based on what has the transition zone been defined as the area within 3 km from human-used areas? I could not find an a priori justification, suggesting that it has been selected based on some goodness-of-fit.*

But this is concerning since, judging by Fig. 2B, very many areas have been excluded, in particular in the cerrado.

- We chose the 3km value based on the study of Broadbent et al. (ref. 25). They found that forest area up to 2km from the forest edge is impacted by logging and the by resulting edge effects up to 2.4km. We took a slightly higher value.
- The Cerrado was also a concern of one of the reviewers in a previous round. According to our data, 44% of the Cerrado lies in the Agricultural/Urban class, 36% in the transition and 20% in the natural class. Hence, $44\%+36\%=80\%$ of the Cerrado was excluded from the analysis of natural cover as it has become heavily impacted by agriculture. However, the Cerrado has been savanna since before large scale human influence. Therefore, the concern was that some of the bimodality would have been removed by wrongly assuming that these savannas are anthropogenic. Nonetheless, when we properly accounted for average rainfall, seasonality and soils, we see that savannas are predicted in the Cerrado without the necessity for human impact (Figure 1B). [In fact, the edge between Amazonian forest and the Cerrado is well predicted with average rainfall and rainfall seasonality alone (Supplementary Figure 4B).] Given that the statistical model predicts the natural forest-savanna front well, any point pattern in the scatterplots that shows overlapping states for a wide range of CEFS indicates that there is another effect than soils and climate affecting tree cover. In transition areas, this is indeed the case, and given that they are situated in areas close to agricultural areas, we conclude that this other effect is human impact. This argument only holds if there is no dependence between false positive prediction of forest and pixels being in the agricultural/urban or transition class. If there is dependence, the overlapping states in the transition class may be due to a poorer representation of natural processes by CEFS. However, there was no significant indication for dependence ($p=0.19$ with a χ^2 -test).

Minor comments

- Low tree cover areas near human-affected areas (the transition zones) are not considered true savannas but rather degraded forests. The authors mention in their reply to referee 1 in the previous round that there exist no large-scale observations about the occurrence of savanna vs. forest trees. However, savanna trees are lower than forest trees, and a recent study (4) related tree cover to remotely sensed canopy height; they found large correspondence between the high tree cover mode and a high canopy height mode, characteristic of forests (and between a low cover mode and low height mode, characteristic of savannas). Nevertheless, they also found that some low cover areas had high canopies, indicating degraded forests. It is plausible that these degraded forests occur relatively often in the human-natural transition zones as defined here (note that this cannot be checked with the coarse-resolution canopy height data), but at least those results are an indication that low tree cover generally rightly captures savanna tree species.

- Thank you for the suggestion. We incorporated this in the revised discussion. We also made an analysis of where these degraded forests occur (with the chosen condition height $> 15\text{m}$ and cover $< 40\%$) and noticed that they indeed occur especially in what we have called the transition zones (see Methods and Supplementary Figure 9).

- In lines 72-73 it is mentioned that this work studies the impact of humans on Amazonian forest-savanna bistability. However, the question how humans affect this bistability is not answered explicitly. It is sufficiently clearly articulated that there exists no bistability without human impact, but do the authors believe that there is bistability with human impact? Or is it just apparently so, because low tree cover in the transition zones is artificial?

- We have clarified the concept of bistability more by distinguishing between local and global bistability, the first being bistability without spatial interaction and the second despite spatial interaction (see also the reply to the editor). The model is locally bistable but not globally bistable. The spatial interaction caused by fire spread always leads to fronts. Hence with *and* without human impact, there is no hysteresis, but the heterogeneity of the Maxwell point introduced by human impact leads to bimodality in the plots. We have made these points now clearer throughout the text and added an explicit conclusion about bistability and human impact (see Discussion).

- It is not without reason that some previous studies did not account for soil variables in their analyses: the quality of the soil databases does not match that of tree cover, so the matter of soil effects cannot be

settled in a conclusive way yet. Using available data is definitely better than nothing, but this limitation should be acknowledged in the manuscript.

- Indeed. We have now acknowledged that. Thank you.

- Lines 259-261: forests might grow more easily in a degraded forest area than in savanna, so a degraded forest does not need to be an alternative stable state to a non-degraded forest; but that does not mean that forest and savanna would not be alternative stable states.

- We see degraded forest as being in transition between forest and savanna. In any case, from the perspective of the fire-vegetation feedback, whether there are savanna or forest trees between the patches of grass does not matter. The alternative stable states are a state of low grass cover and high tree cover ($\sim 80\%$) versus a state of high grass cover and low tree cover ($\sim 0-30\%$).

Literature (cited by referee)

1. Hirota M, Holmgren M, Van Nes EH, & Scheffer M (2011) Global resilience of tropical forest and savanna to critical transitions. *Science* 334(6053):232-235.
2. Staver AC, Archibald S, & Levin SA (2011) The global extent and determinants of savanna and forest as alternative biome states. *Science* 334(6053):230-232.
3. Laurance WF, Ferreira LV, Rankin-de Merona JM, & Laurance SG (1998) Rain forest fragmentation and the dynamics of Amazonian tree communities. *Ecology* 79(6):2032-2040.
4. Xu C, et al. (2016) Remotely sensed canopy height reveals three pantropical ecosystem states. *Ecology* 97(9):2518-2521.

Reviewers' Comments:

Reviewer #1 (Remarks to the Author)

This paper develops the idea that bimodality and apparent hysteresis between forest and savanna in tropical South America is enhanced in areas with more human impact, and less extensive (they argue absent) in areas remote (e.g. > 3 km) from urban areas, agriculture and pasture. In these 'human influenced' zones they suggest that selective (or small scale) logging and fire cause forest cover loss that blurs the forest-savanna boundary. The most interesting point is that, in areas remote from human influences, hysteresis between forest and savanna is reduced or removed by spatial interactions.

The revised manuscript improves somewhat on the previous version. The primary argument, that spatial interactions can reduce hysteresis in systems prone to bifurcation (such as savanna-forest transitions) will be of interest to readers of Nature-Communications. That argument is simple and seems to be demonstrated in the observations and model. The secondary point, that human actions (particularly harvest of trees) will reduce forest cover and increase fires, potentially misleading us to conclude that 'natural' dynamics (i.e. not driven by land use) create considerable hysteresis is also important.

Reviewer #2 (Remarks to the Author)

Mr Wuyts and colleagues have satisfactorily addressed my previous suggestions. I therefore recommend publication, below a few last minor suggestions.

Minor suggestions

Line 15 "bimodality"

Line 97 "Model results further show that there is no hysteresis .. ", the model results show that hysteresis is not required to reproduce the bimodal distribution observed in the data, not necessarily that there is no hysteresis. Maybe rephrase to reflect that the data and model together suggest there is no hysteresis over large scales as previously suggested.

Lines 184 – 205, make sure that your refer to the figures in the correct way (former fig. 2C-E is now fig. 4 A-C).

Figure 4, please consider masking regions of insufficient data in white (for readability). Also, the MODIS vegetation continuous fields data is available through 2013 (MOD44B collection 051). Please mention if collection 5 or 5.1 data has been used and consider using the latest data for this figure. Also, this data is not very suitable for analyzing change, so the authors could also consider to compare mean values (2000-2002) against (2011-2013) or so to characterize change.

Line 265, remove "easily" (?)

Line 306 "The possible responsible processes for the hysteresis are", maybe change to "Possible responsible processes for hysteresis are .. ". The authors could also mention fire – vegetation feedbacks through altered fuel properties.

Lines 326-327 "Naturally bistable forests in .. deterministic savannas", repetition of the previous phrase (?).

Line 349 "howthey"

Line 447, "very high" Scientist in many fields consider a "250 m" resolution to be moderate or coarse.

Reviewer #3 (Remarks to the Author)

I am glad to see the rephrasing of the interpretations of the results, specifically the more nuanced statements about the absence of hysteresis as they were justified by (the current) figure 4. Also, the distinction between local and global bistability and the additional analyses regarding degraded forests have improved the manuscript.

I would like to thank the authors for the figure in their rebuttal that plotted tree cover against CEFS when dry forests are included. As expected, this increased the environmental overlap of high and low tree cover. Whether the exclusion of dry forests is justified is debatable at least. I understand and accept the authors' rationale for not introducing them in the model, but still think excluding "areas where forests may have specific or regionally distinct adaptations to drought, fire or altitude" in the empirical analyses is a very important decision with far-reaching effects on the outcome of the study, and should therefore be mentioned in the discussion. Currently only the above statement is made in the Methods section at the end of the study, but its implications should be more prominently discussed.

The authors refer to Supplementary Figure 6 for the statement that bimodality was only consistent with models in previous studies because these models did not consider spatial interaction. But it is unclear to me whether the authors consider the areas marked as bistable in Supplementary Figure 6D as really bistable based on their own study, or as apparent bistability if the approach by previous studies is taken. In case of the former, I would say that there is still quite extensive bistability.

Some further minor points:

- In the abstract, "human-impacted zones" and "close to human-impacted zones" are currently equated. Be consistent in wording.
- The discussion mentions the Amazon only once, at the end. As it is unclear how generalizable to other areas the results are, mention more clearly the Amazon at the start of the discussion
- "in previous studies ... models did not include spatial interaction" -> there have been models of forest-savanna transitions that did consider spatial interaction, including ones cited.
- "due to bistability and hysteresis on a smaller scale than previously recognised" -> I fail to see how our understanding of the scale of hysteresis has changed. Could this be explained?
- "The possible responsible processes for the hysteresis are: (i) climate variability and extreme events, (ii) fire size distribution effects, and (iii) vegetation-climate feedbacks." -> Of these, only the third can be a process responsible for hysteresis, as hysteresis requires positive feedback. Please revise.
- Here and there, revisions seem to be hastily written down, without sufficient editorial control. Please carefully go through the text again to remove mistakes.
- The reference list is very sloppy. For instance, merely by glancing over the list I could see that at least references 4, 18, 19, 22, 36, 38, 52 and 57 have incomplete bibliographic information. Revisit those papers (as they have appeared in journal issues), and check the entire reference list, to make sure all information is correct.

Detailed reply to Reviewer 1

This paper develops the idea that bimodality and apparent hysteresis between forest and savanna in tropical South America is enhanced in areas with more human impact, and less extensive (they argue absent) in areas remote (e.g. > 3 km) from urban areas, agriculture and pasture. In these 'human influenced' zones they suggest that selective (or small scale) logging and fire cause forest cover loss that blurs the forest-savanna boundary. The most interesting point is that, in areas remote from human influences, hysteresis between forest and savanna is reduced or removed by spatial interactions.

The revised manuscript improves somewhat on the previous version. The primary argument, that spatial interactions can reduce hysteresis in systems prone to bifurcation (such as savanna-forest transitions) will be of interest to readers of Nature-Communications. That argument is simple and seems to be demonstrated in the observations and model. The secondary point, that human actions (particularly harvest of trees) will reduce forest cover and increase fires, potentially misleading us to conclude that 'natural' dynamics (i.e. not driven by land use) create considerable hysteresis is also important.

- Thank you once again for the positive reception. The paper would not have reached the same level without your detailed feedback.

Detailed reply to Reviewer 2

Mr Wuyts and colleagues have satisfactorily addressed my previous suggestions. I therefore recommend publication, below a few last minor suggestions.

- Thank you for your criticisms that have led to a very much improved discussion.

Minor comments

Line 15 “bimodality”

- Thank you very much for spotting this typo. It has been corrected now.

Line 97 “Model results further show that there is no hysteresis .. ”, the model results show that hysteresis is not required to reproduce the bimodal distribution observed in the data, not necessarily that there is no hysteresis. Maybe rephrase to reflect that the data and model together suggest there is no hysteresis over large scales as previously suggested.

- Good point. That is indeed what we intended to convey. We have now rephrased.

Lines 184 – 205, make sure that you refer to the figures in the correct way (former fig. 2C-E is now fig. 4 A-C).

- Thanks again for spotting this. It has been corrected now.

Figure 4, please consider masking regions of insufficient data in white (for readability). Also, the MODIS vegetation continuous fields data is available through 2013 (MOD44B collection 051). Please mention if collection 5 or 5.1 data has been used and consider using the latest data for this figure. Also, this data is not very suitable for analyzing change, so the authors could also consider to compare mean values (2000-2002) against (2011-2013) or so to characterize change.

- After making sure that the colour scale meets the editorial requests, we think that using gray to mask regions with insufficient data is better than white, as white is the central value of the colour scale. This is version 5.1 - we have mentioned it now in the text. The point about not being suited for change was also made by Reviewer 1 and 3 in one of the previous review rounds, then in Nature Geoscience. The following point summarises our reply.
- We were indeed aware of the biologically unjustified fluctuations in the VCF product and did indeed compute a population measure of change - the median of the change in every 2D bin in (CEFS, TC) space, as mentioned in the main article and the supplementary information. We greyed out areas in the plot where the number of observations was small. We have also excluded bad quality pixels (due to cloudiness or other issues). Taking population measures of the change as we did reduces the effect of random errors but does not mitigate systematic errors in the data set (which tend to yield lower estimates). By taking into account the quality flags of the data (as explained in our last submission), we have reduced the systematic errors as much as possible while keeping a sufficiently large sample size.
- We have considered your suggestion of using averages over multiple years previously. However, this choice is not the best one as the sample size over which we average will be limited (three in your case) and averaging over time will blur human impact. Therefore, we chose to compute the change of the median in every bin in (CEFS, TC) space, as mentioned above. The population of pixels from which the population measure is computed (the median here) is much larger in this case (hundreds to thousands).
- We chose the time period of the data as the one that made the most sense for all data combined (rainfall, land cover, tree cover). Changing this will introduce temporal mismatches between data sets.

Line 265, remove “easily” (?)

- We agree. It has been removed now.

Line 306 “The possible responsible processes for the hysteresis are”, maybe change to “Possible responsible processes for hysteresis are .. ”. The authors could also mention fire – vegetation feedbacks through altered fuel properties.

- That is indeed better. We have deleted 'The'. The fire-vegetation feedbacks are implemented in our model by having nonlinearly higher fire occurrence in areas with lower tree cover (see Supplementary Figure 4C-D).

Lines 326-327 “Naturally bistable forests in .. deterministic savannas”, repetition of the previous phrase (?).

- We have now added the word 'Similarly' to make it less repetitive while showing that the discussed process is the same.

Line 349 “howthey”

- Thanks for spotting this. We introduced a space.

Line 447, “very high” Scientist in many fields consider a “250 m” resolution to be moderate or coarse.

- OK. We rephrased to convey just the fact that we have a high number of pixels.

Figure 1: Areas of stability according to Staver et al. (2011). In the Amazon region, bistable areas (dark orange and light green) are much larger than deterministic areas (dark green and light orange). Compare this with Supplementary Figure 6D, where it is the other way round, i.e. deterministic areas take much more space than bistable areas.

Detailed reply to Reviewer 3

I am glad to see the rephrasing of the interpretations of the results, specifically the more nuanced statements about the absence of hysteresis as they were justified by (the current) figure 4. Also, the distinction between local and global bistability and the additional analyses regarding degraded forests have improved the manuscript.

- Many thanks for your response that likewise led to a very much improved discussion.

Major comments

I would like to thank the authors for the figure in their rebuttal that plotted tree cover against CEFS when dry forests are included. As expected, this increased the environmental overlap of high and low tree cover. Whether the exclusion of dry forests is justified is debatable at least. I understand and accept the authors' rationale for not introducing them in the model, but still think excluding "areas where forests may have specific or regionally distinct adaptations to drought, fire or altitude" in the empirical analyses is a very important decision with far-reaching effects on the outcome of the study, and should therefore be mentioned in the discussion. Currently only the above statement is made in the Methods section at the end of the study, but its implications should be more prominently discussed.

- OK. We have now included and discussed this in the discussion.

The authors refer to Supplementary Figure 6 for the statement that bimodality was only consistent with models in previous studies because these models did not consider spatial interaction. But it is unclear to me whether the authors consider the areas marked as bistable in Supplementary Figure 6D as really bistable based on their own study, or as apparent bistability if the approach by previous studies is taken. In case of the former, I would say that there is still quite extensive bistability.

- We indeed meant the former case. When we say "the bimodality as found in previous studies", we refer to bimodality occurring over a large range of the predictor variable. Although our found area of bimodality is not negligible, which is why the last revision was necessary, it is very much reduced compared to previous studies [see Figure 1 in this document, which is Figure 4 of Staver et al. (2011) - ref. 10].

Minor comments

- In the abstract, "human-impacted zones" and "close to human-impacted zones" are currently equated. Be consistent in wording.

- Good point. We have now consistently used "close to".

- *The discussion mentions the Amazon only once, at the end. As it is unclear how generalizable to other areas the results are, mention more clearly the Amazon at the start of the discussion*

- OK. This has been done now.

- *“in previous studies ... models did not include spatial interaction” -> there have been models of forest-savanna transitions that did consider spatial interaction, including ones cited.*

- The only model that we cited and that included spatial interaction, ref. 16, did so via a percolation model. The spatial interaction in this model is different in two ways. (1) Fire is simulated on a smaller scale than we did, with some fires percolating over their whole study area. In our model, fire spread is limited to the neighbouring pixels, agreeing with a range of ~1km. (2) Fires percolate only on pixels with grass and saplings; hence at the forest-savanna boundary, they do not seep into forests, unlike in our model. We have now stressed and nuanced the above statement.

- *“due to bistability and hysteresis on a smaller scale than previously recognised” -> I fail to see how our understanding of the scale of hysteresis has changed. Could this be explained?*

- When bimodality occurs over a wider range of the predictor variable(s), the area that is classified as bistable will be larger. Hence, a limited range of overlap means a limited area in which hysteresis is possible.

- *“The possible responsible processes for the hysteresis are: (i) climate variability and extreme events, (ii) fire size distribution effects, and (iii) vegetation-climate feedbacks.” -> Of these, only the third can be a process responsible for hysteresis, as hysteresis requires positive feedback. Please revise.*

- The reason that we said that these processes may be responsible for hysteresis is that we expected that they may interact with the fire feedback (or other feedbacks) in such ways that may increase the scale of the hysteresis. However, they are indeed not necessarily providing new feedbacks themselves. Thank you for this remark. We have now mentioned interannual climate variability and fire distribution effects (longer-range fire effects) as effects that may cause bimodality due to their stochasticity, and not due to hysteresis. Such stochasticity of tree recruitment and fire occurrence leads to a Maxwell range rather than a specific Maxwell point. Within this range, either forest or savanna is possible, and the position of the forest-savanna front depends on the recent history of fire and climate.

- *Here and there, revisions seem to be hastily written down, without sufficient editorial control. Please carefully go through the text again to remove mistakes.*

- We checked everything more carefully now and have removed typos.

- *The reference list is very sloppy. For instance, merely by glancing over the list I could see that at least references 4, 18, 19, 22, 36, 38, 52 and 57 have incomplete bibliographic information. Revisit those papers (as they have appeared in journal issues), and check the entire reference list, to make sure all information is correct.*

- Thanks for pointing this out. I used EndNote and wrongly assumed the downloaded references were complete. We have now added the missing information.